# ROS/TNF-α Crosstalk Triggers the Expression of IL-8 and MCP-1 in Human Monocytic THP-1 Cells via the NF-κB and ERK1/2 Mediated Signaling

**DOI:** 10.3390/ijms221910519

**Published:** 2021-09-29

**Authors:** Nadeem Akhter, Ajit Wilson, Reeby Thomas, Fatema Al-Rashed, Shihab Kochumon, Areej Al-Roub, Hossein Arefanian, Ashraf Al-Madhoun, Fahd Al-Mulla, Rasheed Ahmad, Sardar Sindhu

**Affiliations:** 1Department of Immunology & Microbiology, Dasman Diabetes Institute, P.O. Box 1180, Dasman 15462, Kuwait; nadeem.akhter@dasmaninstitute.org (N.A.); ajit.wilson@dasmaninstitute.org (A.W.); reeby.thomas@dasmaninstitute.org (R.T.); fatema.alrashed@dasmaninstitute.org (F.A.-R.); shihab.kochumon@dasmaninstitute.org (S.K.); areej.abualroub@dasmaninstitute.org (A.A.-R.); hossein.arefanian@dasmaninstitute.org (H.A.); rasheed.ahmad@dasmaninstitute.org (R.A.); 2Department of Genetics & Bioinformatics, Dasman Diabetes Institute, P.O. Box 1180, Dasman 15462, Kuwait; ashraf.madhoun@dasmaninstitute.org (A.A.-M.); fahd.almulla@dasmaninstitute.org (F.A.-M.); 3Animal & Imaging Core Facility, Dasman Diabetes Institute, P.O. Box 1180, Dasman 15462, Kuwait

**Keywords:** ROS, TNF-α, IL-8, CXCL8, MCP-1, CCL2, oxidative stress, meta-inflammation, obesity

## Abstract

IL-8/MCP-1 act as neutrophil/monocyte chemoattractants, respectively. Oxidative stress emerges as a key player in the pathophysiology of obesity. However, it remains unclear whether the TNF-α/oxidative stress interplay can trigger IL-8/MCP-1 expression and, if so, by which mechanism(s). IL-8/MCP-1 adipose expression was detected in lean, overweight, and obese individuals, 15 each, using immunohistochemistry. To detect the role of reactive oxygen species (ROS)/TNF-α synergy as a chemokine driver, THP-1 cells were stimulated with TNF-α, with/without H_2_O_2_ or hypoxia. Target gene expression was measured by qRT-PCR, proteins by flow cytometry/confocal microscopy, ROS by DCFH-DA assay, and signaling pathways by immunoblotting. IL-8/MCP-1 adipose expression was significantly higher in obese/overweight. Furthermore, IL-8/MCP-1 mRNA/protein was amplified in monocytic cells following stimulation with TNF-α in the presence of H_2_O_2_ or hypoxia (*p* ˂ 0.0001). Synergistic chemokine upregulation was related to the ROS levels, while pre-treatments with NAC suppressed this chemokine elevation (*p* ≤ 0.01). The ROS/TNF-α crosstalk involved upregulation of *CHOP*, *ERN1*, *HIF1A*, and NF-κB/ERK-1,2 mediated signaling. In conclusion, IL-8/MCP-1 adipose expression is elevated in obesity. Mechanistically, ROS/TNF-α crosstalk may drive expression of these chemokines in monocytic cells by inducing ER stress, *HIF1A* stabilization, and signaling via NF-κB/ERK-1,2. NAC had inhibitory effect on oxidative stress-driven IL-8/MCP-1 expression, which may have therapeutic significance regarding meta-inflammation.

## 1. Introduction

Chemotactic cytokines called chemokines play a role in driving the metabolic inflammation associated with obesity, which later leads to the development of type 2 diabetes (T2D), or results in metabolic syndrome. Therefore, it is important to study the factors and mechanisms of expression of obesity-related chemokines, since these are the early pathophysiologic changes that could later lead to more serious metabolic disorders, such as T2D, atherosclerosis, cardiovascular disease, hepatic steatosis, and metabolic syndrome. These small proteins act as “directional signals” which regulate the trafficking of effector leukocytes to the sites of inflammation. The human chemokine system comprises of about 50 chemokines and 20 receptors, belonging to the seven-transmembrane G protein-coupled receptor family [1]. Based on the motif involving two N-terminal cysteines, chemokines are classified as C, CC, CXC, and CX3C types, where “X” represents an amino acid residue [2]. Notably, the CC or β-chemokines usually mediate chronic inflammation through effect on monocytes and T-lymphocytes, whereas the CXC or α-chemokines play a role in acute inflammation through neutrophil chemotaxis [3].

Tumor necrosis factor (TNF)-α, IL-8 (or CXCL8), and MCP-1 (or CCL2) are included among the key cytokine/chemokines that induce and exacerbate metabolic inflammation. It is known that the circulatory levels of IL-8, MCP-1, and TNF-α are significantly upregulated in metabolic disease conditions [4,5,6,7]. Regarding significant functions and expressor cell populations, IL-8 regulates the chemotaxis of neutrophils and is secreted by different cell types, including activated monocytes/macrophages, lymphocytes, adipocytes, keratinocytes, endothelial and epithelial cells [8,9]. Whereas, MCP-1 is a key chemokine involved in regulation, migration and infiltration of monocytes/macrophages as well as memory T cells and natural killer cells; and it is expressed by a wide variety of cells including monocytic, endothelial, epithelial, fibroblasts, smooth muscle, microglial, mesangial, and astrocytic cells [10,11]. While, TNF-α is a prototypic inflammatory cytokine and neutrophil chemoattractant, which is mainly produced by monocytes/macrophages in response to immunological challenges and to a lesser extent, it is produced by other cell types including endothelial cells, lymphoid cells, adipose tissue, cardiomyocytes, mast cells, neurons, and fibroblasts [12]. As a major proinflammatory cytokine involved in early/acute inflammatory events, TNF-α triggers a cascade of several other inflammatory molecules including IL-8, CCL2-4, matrix metallopeptidases, prostaglandins, reactive oxygen species (ROS), and reactive nitrogen intermediates (RNI) [13].

In metabolic disease conditions, increased circulatory levels of IL-8 and MCP-1 in obesity were found to be positively associated with body mass index (BMI), waist circumference, high-density lipoprotein (HDL) cholesterol, IL-6, c-reactive protein (CRP), and homeostatic model assessment of insulin resistance (HOMA-IR) [14]. While, IL-8/TNF-α expression in individuals with obesity/T2D was found to correlate with inflammation and insulin resistance [15,16].

As an early pathophysiological change in the adipose tissue and a co-factor involved in inflammation, oxidative stress plays as a key role in obesity-related co-morbidities [17]. The major contributory factors in metabolic conditions that may lead to oxidative stress following ROS induction include chronic inflammation, hyperglycemia, hyperlipidemia, hyperleptinemia, and impaired endothelial and mitochondrial functions [18,19]. The elevation of ROS, in turn, can further promote the expression of key transcription factors, growth factors, and inflammatory proteins such as TNF-α, IL-1β, IL-6, and CCL2/5 [20].

Previous studies point to the significance of IL-8, MCP-1, and TNF-α in inflammation or in conditions generating ROS. It still remains unclear whether the crosstalk between TNF-α and ROS can modulate the expression of IL-8 and MCP-1 in human monocytic cells. We hypothesized that the cooperativity between ROS and TNF-α could lead to increased expression of IL-8 and MCP-1 in monocytic cells. Herein, first, we present the clinical data showing that obesity is a positive modulator of IL-8 and MCP-1, and next, we demonstrate in vitro that the ROS/TNF-α crosstalk promotes the expression of IL-8 and MCP-1 in human monocytic THP-1 cells through the mechanisms involving endoplasmic reticulum (ER) stress, stabilization of hypoxia-inducible factor (HIF)-1α, and activation of the nuclear factor(NF)-κB and extracellular signal-regulated kinase (ERK)-1/2 signaling pathways. These findings decipher the interrelation of oxidative stress and inflammation for driving IL-8 and MCP-1 expression in an obesity setting.

## 2. Results

### 2.1. Adipose IL-8 and MCP-1 Expression in Lean, Overweight, and Obese Individuals

We first determined the changes in adipose expression of IL-8 and MCP-1 in overweight (BMI: 27.80 ± 1.60 kg/m^2^) and obese individuals (BMI: 35.08 ± 3.73 kg/m^2^) compared to their lean counterparts (BMI: 22.86 ± 1.89 kg/m^2^), 15 each. Their clinico-demographic characteristics are summarized in Appendix A. To this end, as determined by immunohistochemistry (IHC), protein expression of both IL-8 (Figure 1A,B; *p* < 0.0001) and MCP-1 (Figure 1C,D; *p* < 0.0001) was found to be significantly upregulated in overweight and obese individuals as compared to lean individuals. As expected, a direct association was found between the protein and mRNA expression of IL-8 (r = 0.60 *p* = 0.004, Appendix A) and MCP-1 (r = 0.84 *p* < 0.0001, Appendix A). Positive associations were found between systemic CRP levels and adipose protein expression of IL-8 (r = 0.50 *p* = 0.005, Appendix A) and MCP-1 (r = 0.54 *p* = 0.001, Appendix A); as well as between BMI and protein expression of IL-8 (r = 0.84 *p* < 0.0001, Appendix A) and MCP-1 (r = 0.82 *p* < 0.0001, Appendix A).

Consistent with the IHC data, increased protein expression of IL-8 and MCP-1 was further validated by confocal microscopy (Appendix A, respectively). Overall, these data suggest that adipose expression of IL-8 and MCP-1 is significantly upregulated in obesity.

### 2.2. Oxidative Stress Triggers the Expression of IL-8 and MCP-1 in Human Monocytic THP-1 Cells

In obesity, increased proinflammatory responses and oxidative stress contribute to the development of immuno-metabolic complications [21]. We asked whether the co-exposure to oxidative stress and TNF-α could promote the expression of IL-8 and MCP-1 in human monocytic cells. To this end, THP-1 cells were stimulated with TNF-α, in the presence or absence of H_2_O_2_, which is a standard method used to induce ROS-mediated oxidative stress in cells [22,23]. The data show that, as compared to vehicle control, the relative mRNA expression of IL8 was 11-fold increased with TNF-α (*p* < 0.0001) and 4-fold increased with H_2_O_2_ (*p* = 0.002) (Figure 2A); whereas the relative mRNA expression of MCP1 was 7-fold increased with TNF-α (*p* < 0.0001) and 4-fold increased with H_2_O_2_ (*p* = 0.0001) (Figure 2B). However, co-stimulation with TNF-α and H_2_O_2_ cooperatively upregulated the IL8 and MCP1 gene expression by 49-fold and 15-fold, respectively (*p* < 0.0001) (Figure 2A,B).

We further investigated if 1% hypoxia could also induce a similar effect in these cells. To this effect, as expected, we found that 1% hypoxia also elevated the gene expression of IL8 and MCP1 by 3-fold (*p* = 0.02) (Figure 2C) and by 4-fold (*p* < 0.05) (Figure 2D), respectively, as compared to normoxia. However, as expected, TNF-α stimulation under 1% hypoxia, induced significantly high gene expression of both IL8 (23-fold) and MCP1 (70-fold) as compared to normoxia control (*p* < 0.0001) (Figure 2C,D). Since H_2_O_2_ and hypoxia yielded comparable results, we mostly used H_2_O_2_ treatments for inducing oxidative stress in the subsequent experiments since it did not require longer incubation of cells in hypoxia chambers under the complex gaseous mixture containing 1% O_2_, 5% CO_2_, and 94% N_2_.

We next determined IL-8 and MCP-1 protein expression by flow cytometry in THP-1 cells in response to stimulation with TNF-α, in the presence or absence of H_2_O_2_. As shown by flow cytometry images for IL-8 (Figure 3A–C) and MCP-1 (Figure 3D–F), the co-stimulation with TNF-α and H_2_O_2_ induced significantly higher protein expression of IL-8 (MFI: 14194 ± 256, *p* < 0.0001) (Figure 3G) and MCP-1 (MFI: 12,650 ± 176, *p* < 0.0001) (Figure 3H) comparing, respectively, to cell stimulations with either TNF-α (MFI_IL-8_: 6036 ± 132.4, MFI_MCP-1_: 3129 ± 136) or H_2_O_2_ (MFI_IL-8_: 6601 ± 151, MFI_MCP-1_: 3098 ± 201).

Moreover, the elevated protein expression of IL-8 and MCP-1 following co-stimulation of THP-1 cells with TNF-α and H_2_O_2_, as compared to stimulation with either TNF-α or H_2_O_2_, was further confirmed by using confocal microscopy. In corroboration with flow cytometry data, the confocal microscopy also revealed significantly increased expression of IL-8 (Figure 4A,B; *p* < 0.0001) and MCP-1 (Figure 4C,D; *p* = 0.031) in cells that were co-stimulated with TNF-α and H_2_O_2_, as compared to stimulation with TNF-α alone. As expected, strong agreement was found between the transcripts (fold change) and protein expression (MFI) of both IL-8 (r = 0.93, *p* < 0.0001) and MCP-1 (r = 0.96, *p* < 0.0001) in THP-1 monocytic cells (Appendix A). Altogether, these data support that oxidative stress drives the expression of IL-8 and MCP-1 at both the transcriptional and translational levels in human monocytic THP-1 cells.

### 2.3. TNF-α/H_2_O_2_ Cooperativity Triggers ROS, and the ROS Scavenging Suppresses IL-8 and MCP-1 Expression

Next, we determined whether the TNF-α and H_2_O_2_ induced chemokine expression was pivoted to an increase in the intracellular ROS expression, which was measured by dichloro-dihydro-fluorescein diacetate (DCFH-DA) assay. To this end, flow cytometry images (Figure 5A–C) represent a significant increase of the intracelular ROS in the cells that were co-stimulated with TNF-α and H_2_O_2_ (MFI: 31,779 ± 743.60), as compared to those stimulated with either TNF-α (MFI: 1342 ± 51.19) or H_2_O_2_ (MFI: 1650 ± 25.32) (*p* < 0.0001) (Figure 5D). As expected, intracellular ROS had strong positive associations with the gene and protein expression of IL-8 (r = 0.98 and r = 0.91, respectively, *p* < 0.0001) and MCP-1 (r = 0.92 and r = 0.97, respectively, *p* < 0.0001) in THP-1 cells (Appendix A).

To further see if ROS scavenging could suppress the IL8 and MCP1 gene expression, THP-1 cells were treated with an antioxidant N-acetyl cysteine (NAC) or a ROS-inhibitor curcumin, before stimulation with TNF-α and/or H_2_O_2_. To this end, our data show that IL8 gene expression induced by TNF-α and H_2_O_2_ co-stimulation was suppressed significantly by NAC (7-fold, *p* < 0.0001), but not by curcumin (120-fold, *p* = 0.52), as compared to vehicle control (127-fold) (Figure 6A). However, MCP1 gene expression induced by TNF-α and H_2_O_2_ co-stimulation was suppressed significantly (*p* ≤ 0.01) by both NAC (16-fold) and curcumin (23-fold) as compared to vehicle control (56-fold) (Figure 6B). Collectively, these findings present the evidence that ROS-driven IL8 and MCP1 upregulation in THP-1 cells could be effectively counteracted by NAC.

### 2.4. TNF-α/H_2_O_2_ Cooperativity Drives the ER Stress and Stabilizes Hypoxia-Inducible Factor (HIF)-1α

We asked if oxidative stress driving IL-8 and MCP-1 chemokine expression in monocytic cells induced the ER stress and stabilized expression of the oxygen sensor HIF-1α. To this effect, the data show that TNF-α/H_2_O_2_ cooperativity induced significantly higher gene expression of both CHOP (9-fold) and ERN1 (6-fold) in THP-1 cells compared to cell stimulation with H_2_O_2_ alone (CHOP: 6-fold, *p* = 0.007; ERN1: 4-fold, *p* = 0.01) (Figure 7A). A direct regulatory role of ROS in HIF1A expression and stabilization has been reported before [24,25]; however, it remains controversial [26]. Therefore, we next asked whether ROS induction by H_2_O_2_ via a non-canonical mechanism or by 1% hypoxia via a canonical mechanism could both induce the HIF1A gene expression in THP-1 cells. To this effect, a significant increase in HIF1A expression was observed after co-stimulation with TNF-α and H_2_O_2_ (>2-fold), as compared to stimulation with either TNF-α (˂2-fold) or H_2_O_2_ (˂2-fold) (*p* ≤ 0.004, Figure 7B). Similarly, and as expected, stimulation with TNF-α under 1% hypoxia significantly upregulated HIF1A gene expression (40-fold), as compared to stimulation with TNF-α under normoxia (˂2-fold) or exposure to 1% hypoxia alone (˂2-fold) (*p* < 0.0001, Figure 7C). Taken together, these data indicate that THP-1 cell stimulation with TNF-α in the presence of H_2_O_2_ or under 1% hypoxia induces the ER stress and HIF1A stabilization.

### 2.5. Oxidative Stress Promotes the IL-8 and MCP-1 Expression via the NF-κB and ERK1/2 Mediated Signaling

The stress stimuli inducing ROS may activate inflammatory gene expression and we speculated that TNF-α/H_2_O_2_ cooperativity implicated the related signaling pathways including NF-κB and ERK1/2. To this effect, immunoblotting analysis shows increased phosphorylation of NF-κB (9-fold, *p* = 0.0003) and ERK1/2 (6-fold, *p* = 0.03) in THP-1 cells that were co-stimulated with TNF-α and H_2_O_2_, as compared to respective protein expression following stimulation with H_2_O_2_ alone (NF-κB: 3-fold; ERK1/2: 5-fold) (Figure 8A,B, respectively). In agreement with the increased NF-κB phosphorylation, significant degradation of IκBα was observed in the cells that were co-stimulated with TNF-α and H_2_O_2_ (0.6-fold) as compared to stimulation with TNF-α (0.9-fold, *p* = 0.0002) or H_2_O_2_ alone (0.8-fold, *p* = 0.0007) (Figure 8C).

We further demonstrate by flow cytometry that consistent with the Western blot data showing increased NF-κB phosphorylation, there was also an orderly increase in the P-IκBα/IκBα ratio, after THP-1 cell treatments with the vehicle, H_2_O_2_, TNF-α, and TNF-α+H_2_O_2_ (*p* ≤ 0.0001) (Appendix A).

Next, we asked if inhibiting the NF-κB and ERK1/2 mediated signaling could diminish the oxidative stress-induced expression of IL-8 and MCP-1. To this effect, chemical inhibition of NF-κB by nordihydroguaiaretic acid (NDGA) and ERK-1/2 by PD98059 suppressed the phosphorylation of NF-κB (6-fold) (Figure 9A,B, *p* < 0.0001) and ERK-1/2 (6-fold) (Figure 9C,D, *p* < 0.0001) in cells that were co-stimulated with TNF-α and H_2_O_2_ comparing, respectively, to those co-stimulated likewise without pre-incubation with the pathway inhibitors (NF-κB: 10-fold; ERK-1/2: 8-fold).

Furthermore, as expected, protein expression of both IL-8 and MCP-1 was significantly reduced (*p* ≤ 0.001) in THP-1 cells that were co-stimulated with TNF-α and H_2_O_2_, following inhibition of NF-κB by NDGA (MFI_IL-8_: 3755 ± 96; MFI_MCP-1_: 497 ± 2.89) (Figure 10A–H,Q–R) and inhibition of ERK1/2 by PD98059 (MFI_IL-8_: 1085 ± 52.70; MFI_MCP-1_: 435 ± 16.50) (Figure 10I–P,S–T) mediated signaling as compared with IL-8/MCP-1 expression in cells stimulated likewise without pre-incubation with NDGA (MFI_IL-8_: 8874 ± 88; MFI_MCP-1_: 898 ± 13.80) or PD98059 (MFI_IL-8_: 4180 ± 50.13; MFI_MCP-1_: 953 ± 6.35), confirming that NF-κB and ERK1/2 mediated signaling mainly regulated the oxidative stress-driven expression of IL-8 and MCP-1 in THP-1 monocytic cells.

Overall, our data suggest a model of the IL-8 and MCP-1 amplification by ROS/TNF-α crosstalk in THP-1 monocytic cells, as illustrated in the schematic (Figure 11).

## 3. Discussion

The mechanisms addressing inflammatory gene reprogramming under oxidative stress remain poorly understood. Chronic low-grade inflammation called metabolic inflammation and adipose oxidative stress are the hallmarks of obesity. Increasing evidence points to a potential link between inflammation and oxidative stress in metabolic conditions. In this study, we present the evidence, for the first time to our knowledge, that ROS/TNF-α crosstalk drives the expression of IL-8 and MCP-1 in monocytic cells.

First, we present the clinical evidence that α/β chemokines (IL-8 and MCP-1) adipose expression is significantly high in overweight and obese individuals as compared to their lean counterparts. In obesity, cellular changes in the adipose compartment occur, which include M1 polarization and colonization of adipose tissue by proinflammatory M1 macrophages. These effectors secrete the proinflammatory cytokines/ chemokines and adipokines, which are pivotal to sustaining an inflammatory milieu and inducing insulin resistance [27]. Consistent with our findings of the elevated IL-8/MCP-1 expression in obese states and its association with the systemic inflammation (CRP), Kim et al. found the increased circulating levels of IL-8 and MCP-1 and their positive association with obesity-related parameters including BMI, HOMA, and CRP [14].

Since the increased TNF-α expression and oxidative stress are two primal level pathophysiological triggers in obesity [28,29,30,31], it was intriguing to further explore whether these two agents could modulate IL-8 and MCP-1 expression in monocytic cells. Using THP-1 monocytic cells experimental model, we next showed that ROS/TNF-α crosstalk promotes the expression of IL-8 and MCP-1 in monocytic cells. In this model, the cells were treated with H_2_O_2_ to induce oxidative stress. This agent is commonly employed to study the redox-regulated processes as it is relatively stable intracellularly as compared to other ROS [32]. IL-8 and MCP-1 mainly act as neutrophil and monocyte chemoattractants, respectively. As recruited to site(s) of inflammation, the innate phagocytes such as neutrophils and monocytes/macrophages readily produce the effector molecules such as cytokines, chemokines, granular proteins, and oxidant molecules. Our data showing IL-8 and MCP-1 expression in monocytic cells after stimulation with TNF-α or H_2_O_2_ are partially corroborated by other reports reporting expression of these chemokines in other perspectives, such as IL-8 production and regulation in human adipose tissue shown by Bruun et al. and TNFα-induced *MCP1* gene expression in endothelial cells reported by Chen et al. [33,34].

Importantly, in regard to the novelty aspects of the study, we report the cooperativity between the proinflammatory (TNF-α) and oxidative stress (H_2_O_2_) stimuli, leading to the amplification of these two chemokines in THP-1 cells. We observed a similar amplification pattern of IL-8 and MCP-1 co-expression as the oxidative stress was induced by incubating cells under 1% hypoxia. Nonetheless, comparing levels of transcripts expression, TNF-α/H_2_O_2_ interaction induced more *IL8* and TNF-α/hypoxia interaction induced more *MCP1* expression in THP-1 cells. This differential expression of *IL8* and *MCP1* in monocytic cells by TNF-α, in the presence of ROS induction by non-canonical and canonical mechanisms, is an interesting observation and merits further studies involving different cell types and proinflammatory agents other than TNF-α.

Our data show that THP-1 cell co-stimulation with TNF-α and H_2_O_2_ induces increased intracellular ROS synthesis as compared to stimulation with either TNF-α or H_2_O_2_. Indeed, the inflammatory and oxidative stress processes are inter-twined. Upon activation, many immune cells generate free radicals, which, in turn, promote the ROS synthesis and induce inflammation [35]. TNF-α binding and signaling through its receptor activates the inflammatory signaling pathways and ROS synthesis [36]. TNF-α promotes ROS by facilitating electron transfer to oxygen and thereby generating superoxide anions [37]. In monocytes, several sources may contribute ROS, such as mitochondria, NADPH, and xanthine oxidase [38]. To maintain the cellular redox homeostasis, H_2_O_2_, which is the main component of ROS, is normally scavenged by antioxidant enzymes such as superoxide dismutase, catalase, and glutathione peroxidase. However, during in vitro stimulation, exogenous H_2_O_2_ diffuses through cell membranes in a process regulated by membrane lipids and channel proteins [39], rendering ROS synthesis through the Fenton reaction [40]. ROS, in turn, induces the expression of a wide variety of molecules including inflammatory mediators, adhesion molecules, and growth factors through activation of the redox-sensitive NF-κB/NOX pathways [41]. Our results showing ROS elevation by TNF-α or H_2_O_2_ are in line with other reports [32,42].

Moreover, we found strong associations of ROS with the induction of *IL8* (r = 0.98) and *MCP1* (r = 0.92) gene expression (*p* < 0.0001). To verify the ROS-dependence of the expression of both these chemokines, cells were treated with two ROS scavengers or inhibitors including NAC and curcumin before exposure to TNF-α and/or H_2_O_2_. The cooperative expression of *IL8* was effectively suppressed by NAC and that of *MCP1* was suppressed by both NAC and curcumin. NAC is a prodrug to L-cysteine, which is a precursor of GSH, and it acts as a free radical scavenger by replenishing protective levels of GSH and by producing sulfhydryl groups that directly eliminate ROS [43]. Curcumin is the primary active agent in turmeric, *Curcuma longa Linn*, and is known to have phytonutrient and bioprotective properties [44]. Curcumin inhibits ROS by inhibiting protein kinase C activity and Ca^2+^ influx [45]. Our data showing effective suppression of *IL8* and *MCP1* by NAC and/or curcumin are consistent with the available literature [46,47,48,49].

Next, our data show that H_2_O_2_ treatment of THP-1 cells leads to the increased expression of ER stress markers *CHOP* and *ERN1*, which is further enhanced manifold following co-stimulation with TNF-α and H_2_O_2_. ER is the primary site for protein folding and transportation, and it plays a key role in the regulation of intracellular Ca^2+^ and the synthesis of secretory proteins. Unfolded protein accumulation and intracellular Ca^2+^ depletion cause ER stress, resulting in ROS accumulation, mitochondrial dysfunction, and proinflammatory cytokine expression [50]. Consistent with our results showing the increased expression of *CHOP*/*ERN1* and *IL8*, Vij et al. showed that CHOP regulated IL-8 promoter activity and cyclooxygenase (COX)-2 inhibition led to *CHOP* and *IL8* suppression [51]. Interestingly, Kodama et al. [52] demonstrated that the increase in *CHOP* expression reduced the *MCP1* promoter activity induced by PI3-kinase, while the overexpression of *CHOP* alone upregulated *MCP1* promoter activity in a dose-dependent manner, the latter corroborating the increased *CHOP* and *MCP-1* expression in our experimental model. ERN1, also called IRE1α, is the most prominent and highly conserved ER stress sensing protein and immune cells with strong secretory function have the increased ERN1 activity [53]. In line with our findings of the increased *ERN1* and *MCP1* expression in THP-1 cells, Tufanli et al. showed that ERN1 inhibition led to downregulation of cytokines/chemokines genes in monocytes/macrophages [54]. Our results showing elevated *ERN1* and *IL8* expression in monocytic cells are in line with the study reporting IRE1-dependent proinflammatory signaling, leading to increased expression of proinflammatory chemokines [55].

HIF-1 is an oxygen-regulated transcriptional activator that coordinates gene expression during hypoxia. In addition to upregulation by hypoxia, our data show the non-hypoxic upregulation of *HIF1A* by H_2_O_2_, which is in partial agreement with another report demonstrating HIF-1α stabilization by exogenous H_2_O_2_ [56]. We also found that TNF-α stimulation under hypoxia was a much potent trigger of *HIF1A*, which conforms in principle with the report linking HIF-1α as a key molecule with the TNF-α expression in macrophages under hypoxia [57]. Our findings of HIF1A stabilization together with IL-8 and MCP-1 expression under hypoxic stress are in line, at least partially, with other reports [58,59].

Next, we show that expression of IL-8 and MCP-1 in THP-1 monocytic cells co-stimulated with TNF-α and H_2_O_2_ involves increased phosphorylation of NF-κB and ERK1/2. As expected, increased NF-κB phosphorylation in cells was paralleled with the IκBα degradation as well as an increase in the phospho-IκBα/ total-IκBα ratio. These observations are in agreement with other studies of chemokine expression in different cell types or their induction by different stress stimuli [60,61,62]. Elliott et al. showed that NF-κB binding site was essential for the up-regulation of *IL8* gene expression in different uterine cell types [63]. Other studies also support NF-κB binding to the IL-8 or MCP-1 promoter [64,65,66,67]. Of note, we also found that IL-8 and MCP-1 expression was significantly diminished as the cells were pre-incubated with NDGA or PD98059 to block signaling via the NF-κB or ERK-1/2, respectively, which indicates that these pathways may be critical to the oxidative stress-induced expression of IL-8 and MCP-1 in monocytic cells.

Nonetheless, our study involves certain caveats that warrant caution while interpreting the results. First, the experimental data were obtained using THP-1 monocytic cell line model. Second, the clinical data represent the subcutaneous adipose tissue changes and, therefore, further studies will be required to validate these findings in primary monocytes/macrophages, as well as in visceral adipocytes, together with determining changes in proinflammatory cytokines/chemokines other than IL-8 and MCP-1.

In conclusion, first, we present the clinical evidence that adipose expression of IL-8 and MCP-1 is elevated in individuals with obesity. Next, the experimental in vitro data support that ROS/TNF-α crosstalk triggers the expression of IL-8 and MCP-1 in THP-1 monocytic cells through the mechanisms involving ER stress, *HIF1A* stabilization, and NF-κB/ERK1-2 mediated signaling. Importantly, we also show that NAC has an inhibitory effect on oxidative stress-driven IL-8 and MCP-1 expression, which may have therapeutic significance in meta-inflammation.

## 4. Materials and Methods

### 4.1. Participants, Anthropometry, and Collection of Subcutaneous Adipose Tissue Samples

To determine the adipose tissue expression of IL-8 and MCP-1, 45 individuals were recruited in this study, classified based on their BMI (ratio of body weight in kg to height in m^2^) [68] as lean (22.86 ± 1.89 kg/m^2^), overweight (27.80 ± 1.60 kg/m^2^) and obese (35.08 ± 3.73 kg/m^2^), 15 individuals per group. In accord with the ethical guidelines of the Declaration of Helsinki and approval by the ethics committee of Dasman Diabetes Institute, Kuwait (Protocol #: RA 2010-003); written informed consent was obtained from each participant. Waist circumference was measured using constant tension tape; height was measured using portable inflexible height measuring bars and weight was measured by using calibrated portable electronic weighing scales.

Adipose tissue biopsies (~0.5 g) were collected from the abdominal subcutaneous fat pad lateral to the umbilicus using the standard sterile technique. Briefly, periumbilical area was disinfected using ethanol swab and locally anesthetized using 2% lidocaine (2 mL, Fresenius Kabi, LLC, Lake Zurich, IL, USA). Fat tissue was collected making a small 0.5 cm superficial skin incision, incised into smaller pieces (~50–100 mg), put in RNAlater (Sigma-Aldrich Chemie GmbH, Taufkirchen, Germany) and stored at −80 °C until use.

### 4.2. Immunohistochemistry (IHC)

Paraffin-embedded adipose tissue sections (4 μm thick) were deparaffinized in xylene and rehydrated using descending grades of ethanol (100%, 95%, and 75%) to water. Antigen retrieval was carried out using target retrieval solution (pH 6.0; Dako, Glostrup, Denmark) by pressure cooker boiling for 8 min and cooling for 15 min. After PBS washing, endogenous peroxidase activity was blocked with 3% H_2_O_2_ for 30 min and non-specific antibody binding was blocked with 5% nonfat milk (1 h), followed by 1% bovine serum albumin (BSA) solution (1 h). The samples were incubated overnight at room temperature using primary rabbit polyclonal antibodies against IL-8 (1:200 dilution, Abcam^®^ ab106350, pH 6.0, Cambridge, MA, USA) and MCP-1 (1:400 dilution, Abcam^®^ ab9669, pH 6.0). After washing with PBS (0.5% Tween), samples were incubated for 1 h with secondary, horseradish peroxidase (HRP)-conjugated goat anti-rabbit antibody (EnVision Kit, Dako, Glostrup, Denmark) and color was developed using chromogenic 3,3′-diaminobenzidine (DAB) substrate. Samples were washed in running tap water, lightly counterstained with Harris hematoxylin, dehydrated using ascending grades of ethanol (75%, 95%, and 100%), cleared in xylene, and mounted in dibutylphthalate xylene (DPX). For analysis, digital photomicrographs of adipose tissue sections [20×; PanoramicScan II, 3DHistech, Hungary. URL: https://www.3dhistech.com/products-and-software/hardware/pannoramic-digital-slide-scanners/pannoramic-scan-2/ (accessed on 10 August 2021)] were used to quantify the staining in ten different regions and assess the regional heterogeneity in the tissue samples. The regions were outlined using Aperio ImageScope software [Aperio Vista, CA, USA. URL: https://aperio-imagescope.software.informer.com/9.0/ (accessed on 10 August 2021)]. Aperio-positive pixel count algorithm (version 9) integrated into Imagescope Software was used to quantify the intensity of specific staining in the region. The number of positive pixels was normalized to the number of total pixels (positive and negative) to account for variations in the size of the region sampled. Color and intensity thresholds were set to detect the immuno-staining as positive and the background as negative pixels. Once set, all slides were analyzed using the same parameters. The resultant color markup of the analysis was confirmed for each slide.

### 4.3. Cell Cultures and Treatments

THP-1 human monocytic leukemia cell line was purchased from the American Type Culture Collection and propagated in RPMI-1640 culture medium (Gibco, Life Technologies, Grand Island, USA) containing 10% fetal bovine serum (FBS; Gibco, Life Technologies, Grand Island, NY, USA), 2 mM glutamine (Gibco, Invitrogen, Grand Island, NY, USA), 1 mM sodium pyruvate, 10 mM HEPES, 50 U/mL penicillin and 50 μg/mL streptomycin (Gibco, Invitrogen, Grand Island, NY, USA). Cells (10^6^ cells/mL) were treated with TNF-α (10 ng/mL) and/or H_2_O_2_ (10 mM) and incubated under 5% CO_2_ at 37 °C for 24 h, unless otherwise stated. Cells were collected, harvested by centrifugation, and lysed in RLT buffer (cat# 1015762, Qiagen, GmbH, Germany) for total RNA extraction (RNeasy kit, Qiagen, Valencia, CA, USA) and in cell lysis buffer (cat# 9803, Cell Signaling Technology, MA, USA) for preparing protein lysate, following the manufacturers’ instructions.

For culturing cells under hypoxia (1% O_2_), 10^6^ cells per mL were seeded in triplicate wells of 12-well culture plates and treated with TNF-α (10 ng/mL) in designated wells. To induce hypoxic stress, plates were kept in a hypoxia chamber (StemCell Technologies) and purged with a gaseous mixture containing 1% O_2_, 5% CO_2_, and 94% N_2_ for 20 min (at a flow rate of 10 L/min) before sealing. The sealed chamber was placed in an incubator at 37 °C for 36 h. For normoxia control in parallel, similarly prepared culture plates were incubated at 37 °C under 5% CO_2_ (i.e., in 20% O_2_) for 36 h. Cells were collected, harvested by centrifugation, and lysed in RLT buffer (cat# 1015762, Qiagen, GmbH, Germany) for total RNA extraction using RNeasy kit (Qiagen, Valencia, CA, USA), following the manufacturers’ instructions.

In assays including ROS scavengers/antioxidants, 10^6^ cells per mL in triplicate wells were pre-incubated for 30 min with NAC (1 mM) or curcumin (10 μM) in designated wells before incubating with TNF-α (10 ng/mL) and/or H_2_O_2_ (10 mM) at 37 °C for 2 h. Cells were collected, harvested by centrifugation, and lysed in RLT buffer (cat# 1015762, Qiagen, GmbH, Germany) for total RNA extraction using RNeasy kit (Qiagen, Valencia, CA, USA), following the manufacturers’ instructions.

In assays involving pathway inhibitors, 10^6^ cells per mL in triplicate wells were pre-incubated for 1 h with NF-κB inhibitor NDGA (30 μM) and ERK1/2 inhibitor PD98059 (10 μM), followed by stimulation with TNF-α (10 ng/mL) for 10 min and/or H_2_O_2_ (10 mM) for 30 min. Cell lysates were resolved by 12% SDS-PAGE and blots were probed with primary and secondary antibodies for detection of phosphorylated and total NF-κB (65 KDa) and ERK1/2 (42, 44 KDa) by Western blotting. In similar experiments, THP-1 cells were pre-incubated with the pathway inhibitors as above, followed by stimulation with TNF-α (10 ng/mL) and/or H_2_O_2_ (10 mM) for 24 h, while controls were treated with vehicle only. IL-8 and MCP-1 protein expression (MFI) was measured using flow cytometry in cells with and without inhibitor treatment.

### 4.4. Quantitative, Real-Time Reverse-Transcription Polymerase Chain Reaction (qRT-PCR)

Total RNA was extracted using RNeasy kit (Qiagen, Valencia, CA, USA) and following the manufacturer’s instructions. RNA was quantified using Epoch™ Spectrophotometer System (BioTek, Winooski, USA). Each RNA sample (1 μg) was reverse transcribed into cDNA using random hexamer primers and TaqMan reverse transcription reagents (High-Capacity cDNA Reverse Transcription Kit; Applied Biosystems, CA, USA). For real-time RT-PCR, each cDNA sample (50 ng) was amplified using TaqMan^®^ Gene Expression MasterMix (Applied Biosystems, CA, USA) and target gene-specific TaqMan Gene Expression Assay (Applied Biosystems, CA, USA) products including IL-8 (Hs00174103_m1), MCP-1 (Hs00234140_m1), CHOP (Hs00358796_g1), ERN1 (Hs00980095_m1) and GAPDH (Hs03929097_g1) containing forward and reverse gene-specific primers and a target-specific TaqMan^®^ MGB probe labeled with FAM dye at 5′-end and NFQ-MGB at 3′-end of the probe, with 40 cycles of PCR amplification using TaqMan^®^ Gene Expression Master Mix (Applied Biosystems, Foster city, CA, USA) in 7500 Fast Real-Time PCR System (Applied Biosystems, CA, USA). Each cycle included denaturation (95 °C for 15 s) and annealing/extension (60 °C for 1 min), after uracil DNA glycosylase (UDG) (50 °C for 2 min) and AmpliTaq Gold enzyme (95 °C for 10 min) activation. Target gene expression relative to control was calculated using the comparative 2^−ΔΔCT^ method. Results were normalized to GAPDH expression and expressed as fold change over average control gene expression taken as 1.

### 4.5. Flow Cytometry

THP-1 cells, suspended as 10^6^ cells per mL, were incubated at 37 °C with BSA as the vehicle control as well as TNF-α (10 ng/mL) and/or H_2_O_2_ (10 mM/mL) for 24 h. Brefeldin A was added during the last 6h of incubation. Later, cells were harvested by centrifugation, washed twice with permeabilization buffer, and then stained by incubating separately with PE-conjugated mouse anti-human IL-8 (cat# 554720, BD Biosciences, San Jose, CA, USA) and MCP-1 (cat# 554666, BD Biosciences, San Jose, CA, USA) antibodies for 20 min in a fixation and permeabilization solution (cat# 554714, BD Cytofix/Cytoperm BD biosciences, San Jose, CA, USA). Cells were then washed with permeabilization buffer and resuspended in PBS supplemented with 2% paraformaldehyde for FACS analysis (FACSCanto II; BD Bioscience, San Jose, USA). MFI data were analyzed using BD FACSDiva^TM^ Software 8 (BD Biosciences, San Jose, CA, USA).

### 4.6. Confocal Microscopy

Cultured THP-1 cells were harvested from treatment and control wells (in triplicate), fixed with 4% formaldehyde for 10 min and washed thrice with PBS. Cells were permeabilized using 0.25% Triton X-100 in PBS for 10 min and then incubated with BSA for 1 h. After blocking, cells were incubated with primary rabbit polyclonal antibodies against IL-8 (1:100 dilution, Abcam^®^ ab106350, pH 6.0, Cambridge, MA, USA) and MCP-1 (1:100 dilution, Abcam^®^ ab9669, pH 6.0, Cambridge, MA, USA) at room temperature for 1 h. Cells were washed 3 times using PBS-Tris and incubated for 1 h at room temperature with goat anti-rabbit secondary antibody conjugated with Alexa Fluor-488 (Abcam^®^ ab150077) or with goat anti-rabbit secondary antibody conjugated with Alexa Fluor-647 (Abcam^®^ ab150079). The nuclear counterstaining was performed by curing specimens with 4′,6-diamidino-2-phenylindole (DAPI) (cat# H1500, Vectashield hard-set mounting medium, Vector Laboratories Inc, Burlingame, CA, USA) at room temperature for 15 min and hardened completely at 4 °C overnight. Confocal images were obtained using inverted Zeiss LSM710 Spectral confocal microscope (Carl Zeiss, Gottingen, Germany) and EC Plan-Neofluar 40×/1.30 oil DIC M27 objective lens. After sample excitation using 405 nm and 488 nm lines of an argon-ion laser, optimized emission detection bandwidths were configured using Zeiss Zen 2010 control software. For IL-8 and MCP-1 protein quantification, 10 random images (40× magnification) of stained cells were captured for each treatment. The fluorescent channels were split using Image J software (NIH, USA). The fluorescence intensity for respective markers i.e., IL-8 and MCP-1 was detected in 10 randomly selected fields from the entire tissue section (at 40× magnification) and subtracted from the background of the image. Mean integrated density was calculated for each image.

For measuring IL-8 protein expression in activated monocytes/macrophages (M1) in adipose tissue, after antigen retrieval and blocking, samples were incubated for 2h with anti-human CD80 mouse mAb (1:200 dilution, MA5-15512, ThermoFisher Scientific). After washing, samples were incubated overnight at RT with rabbit polyclonal anti-human IL-8 primary Ab (1:200 dilution, ab106350, Abcam). After 3 washes, samples were incubated for 1 h with secondary Abs; first with AF488-conjugated goat anti-mouse (1:400 dilution, ab150113, Abcam), and after 3 washes with AF647-conjugated goat anti-rabbit (1:400 dilution, ab150079, Abcam). Samples were counterstained with DAPI, mounted and analyzed. For measuring MCP-1 expression, after antigen retrieval and blocking, samples were incubated for 2 h with anti-hu CD80 mouse mAb (1:200 dilution, MA5-15512, ThermoFisher Scientific). After washing, samples were incubated overnight at RT with rabbit polyclonal anti-human MCP-1 primary Ab (1:100 dilution, PA5-80413, ThermoFisher Scientific). After 3 washes, samples were incubated for 1 h with secondary Abs; first with AF647-conjugated goat anti-mouse (1:400 dilution, ab150115, Abcam), and after 3 washes with AF488-conjugated goat anti-rabbit (1:400 dilution, ab150077, Abcam). Samples were counterstained with DAPI, mounted, and analyzed. IL-8 and MCP-1 fluorescence intensities were detected in 10 randomly selected fields of each image and mean integrated density was calculated for each image.

### 4.7. Dichloro-Dihydro-Fluorescein Diacetate (DCFH-DA) Assay for ROS Measurement

After inducing oxidative stress by H_2_O_2_ treatment, intracellular ROS was measured in THP-1 cells using commercial ROS assay kit (cat# KP-06-003 BQC Kit, BioQueChem Inc, Llanera-Asturias, Spain), based on the uptake of cell permeant fluorogenic probe 2′-7′dichlorofluorescein diacetate (DCFH-DA). Following cell incubation with the probe for 15 min, DCFH-DA was hydrolyzed (deacetylated) by cellular esterases into DCFH carboxylate anion, which was then oxidized by ROS into the fluorescent product 2′-7′dichlorofluorescein (DCF), measurable by flow cytometry. For DCFH-DA assay, 10^6^ cells per mL were treated with vehicle (Ctrl), TNF-α (10 ng/mL) and/or H_2_O_2_ (10 mM) for 24 h at 37 °C. Later, cells were stained in culture media using DCFH-DA probe (15 μM) for 30 min at 37 °C and analyzed, without washing, by flow cytometry. The final product (DCF) was excited using 488 nm laser and detected at 535 nm.

### 4.8. Western Blotting

THP-1 monocytic cells, cultured in RPMI-1640 complete medium at a cell density of 10^6^ cells per mL in triplicate wells, were treated with H_2_O_2_ (10 mM) for 30 min and/or TNF-α (10 ng/mL) for 10 min, unless stated otherwise, in a humidified incubator (5% CO_2_), while BSA-treated cells served as vehicle control. Harvested cells were incubated for 30 min with cell lysis buffer containing Tris 62.5 mM (pH 7.5), 1% Triton X-100, and 10% glycerol, lysates were clarified by centrifugation at 10,000× *g* for 10 min and supernatants were collected. Protein concentration was measured using Quickstart Bradford Dye (cat# 5000205, Bio-Rad Laboratories, Hercules, CA, USA). Cell lysates were resolved using 12% SDS-PAGE. Blots were probed with rabbit anti-human antibodies (1:1000 dilution) against phospho-NF-κB, total-NF-κB, phospho-ERK1/2, total-ERK1/2, phospho-IκBα, total-IκBα and beta actin at 4 °C overnight. All primary antibodies were purchased from Cell Signaling (CST Inc., Danvers, MA, USA). Blots were washed with TBS 3 times and incubated with HRP-conjugated secondary antibody (1:2500 dilution, Promega, Madison, WI, USA) for 2 h. Immunoreactive bands were developed by using Amersham ECL Plus Western Blot Detection System (GE Health Care, Buckinghamshire, UK) and visualized by ImageDoc™ MP Imaging Systems (Bio-Rad Laboratories, Hercules, CA, USA).

### 4.9. Statistical Analysis

The data obtained were expressed as mean ± standard error of mean (SEM) values and group means were compared using one-way ANOVA (Tukey’s multiple comparisons test) or two-way ANOVA, as required. GraphPad prism software (Version 9.0.2.161, San Diego, CA, USA) was used for graph preparation and statistical analysis. For all analyses, *p*-values ˂ 0.05 were considered statistically significant.

## Figures and Tables

**Figure 1 ijms-22-10519-f001:**
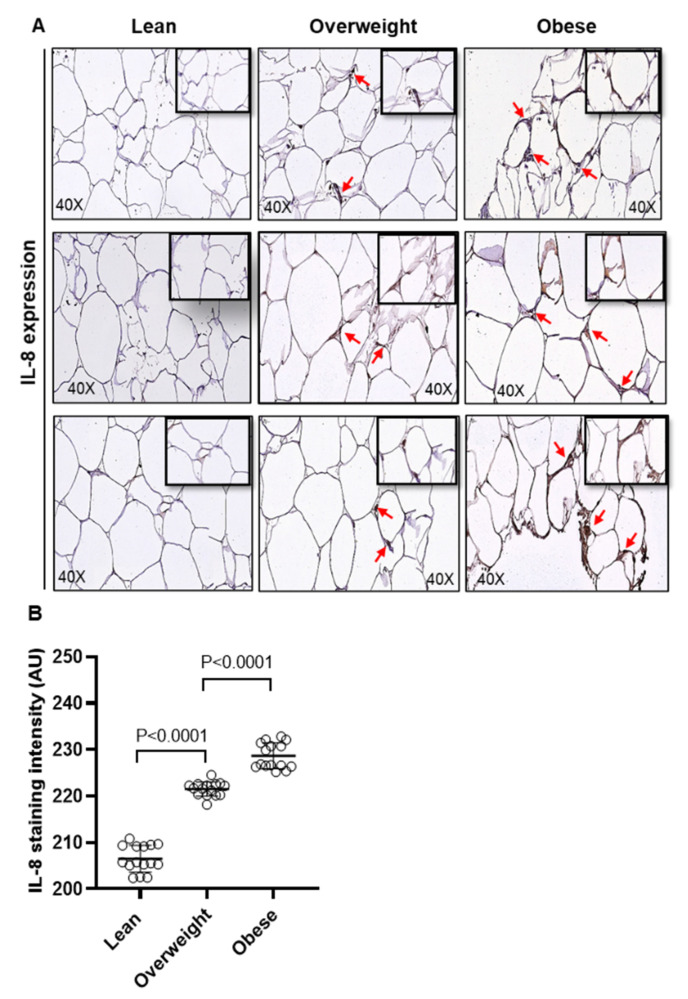
Adipose IL-8 and MCP-1 expression in lean, overweight, and obese individuals. IL-8 and MCP-1 protein expression was determined in adipose tissue samples from lean (BMI: 22.86 ± 1.89 kg/m^2^), overweight (BMI: 27.80 ± 1.60 kg/m^2^), and obese (BMI: 35.08 ± 3.73 kg/m^2^), 15 individuals per group, using immunohistochemistry (IHC) as described in the materials and methods. One-way ANOVA (Tukey’s multiple comparisons test) was used to calculate group differences and *p*-values less than 0.05 were considered as significant. (**A**) Representative IHC staining performed in triplicate shows the elevated IL-8 protein expression (red arrows, 40× magnification) in obese and overweight as compared to lean, depicted for 3 of 15 individuals in each group. (**B**) IL-8 expression intensity assessed in 10 fields per sample, shown as arbitrary units (AU), was found to be significantly higher in obese and overweight individuals, as compared to their lean counterparts (*p* < 0.0001). (**C**) Representative IHC staining (in triplicate), shows the elevated MCP-1 protein expression (red arrows, 40× magnification) in obese and overweight as compared to lean, depicted for 3 of 15 individuals in each group. (**D**) MCP-1 expression intensity assessed in 10 fields per sample, shown as arbitrary units (AU), was found to be significantly higher in obese and overweight individuals, as compared to lean controls (*p* < 0.0001).

**Figure 2 ijms-22-10519-f002:**
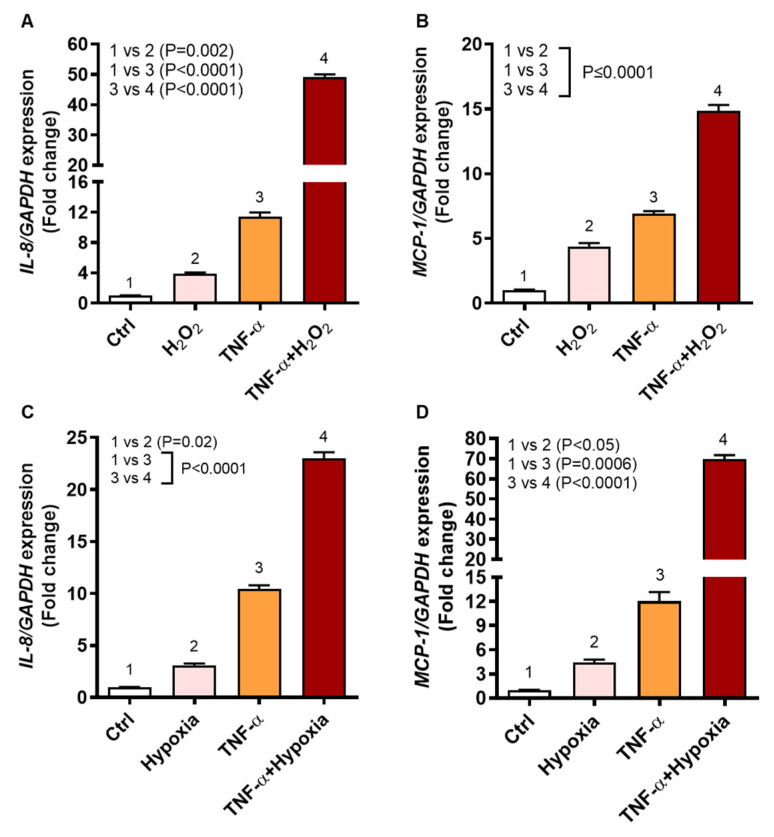
IL8 and MCP1 gene expression in human monocytic THP-1 cells stimulated with TNF-α and/or oxidative stress. THP-1 cells were stimulated, in triplicate, with TNF-α (10 ng/mL) and/or H_2_O_2_ (10 mM) for 24 h while control cells were treated with vehicle only. Likewise, THP-1 cells were also stimulated, in triplicate, with TNF-α under oxidative stress by 1% hypoxia. Total RNA was collected from cells for measuring gene expression of IL8 and MCP1 using qRT-PCR as detailed in methods. One-way ANOVA (Tukey’s multiple comparisons test) was used to calculate group differences and *p*-values less than 0.05 were considered as significant. The representative data (mean ± SEM) obtained from three independent experiments with similar results show the upregulated transcripts of (**A**) IL8 and (**B**) MCP1 in cells co-stimulated with TNF-α and H_2_O_2_, compared to TNF-α stimulation alone (*p* ≤ 0.0001). Similarly, the representative data (mean ± SEM) from three independent experiments with similar results show increased transcripts expression of (**C**) IL8 and (**D**) MCP1 in the cells stimulated with TNF-α under 1% hypoxia, compared to TNF-α stimulation under normoxia (20% O_2_) (*p* < 0.0001).

**Figure 3 ijms-22-10519-f003:**
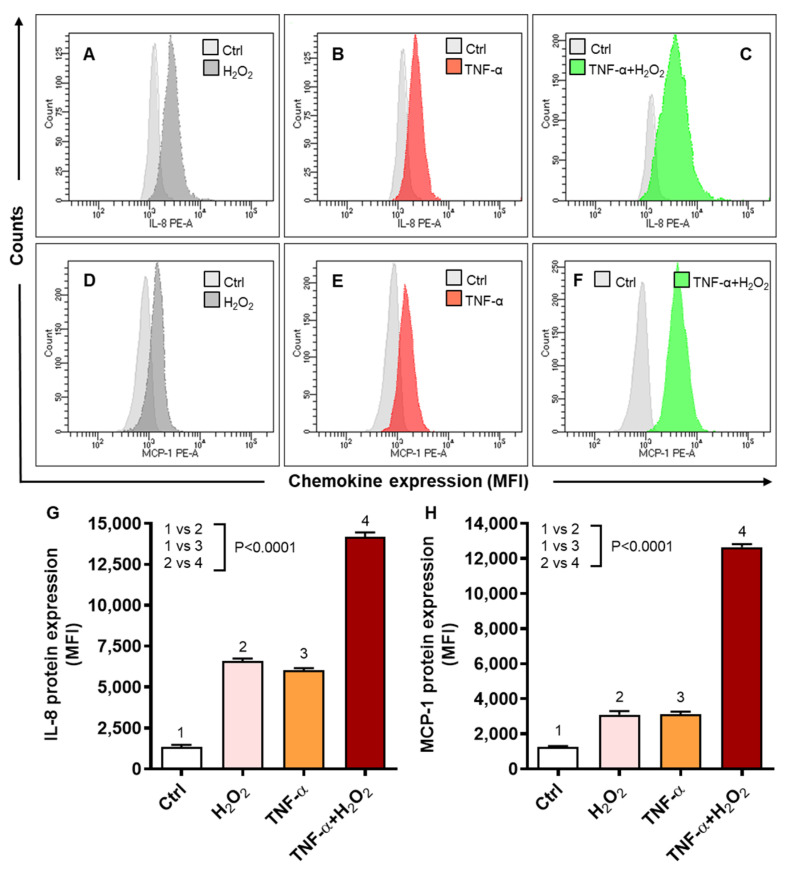
IL-8 and MCP-1 protein expression in THP-1 cells stimulated with TNF-α in the presence of oxidative stress (H_2_O_2_). THP-1 cells were stimulated, in triplicate, with TNF-α (10 ng/mL) and/or H_2_O_2_ (10 mM) for 24 h, while control cells were treated with vehicle only. IL-8 and MCP-1 protein expression was measured using flow cytometry as described in materials and methods. One-way ANOVA (Tukey’s multiple comparisons test) was used to calculate group differences and *p*-values less than 0.05 were considered as significant. The representative histograms from three independent experiments with similar results show (**A**–**C**) expression of IL-8 and (**D**–**F**) expression of MCP-1 in cells stimulated with TNF-α and/or H_2_O_2_ as compared to control. Quantitative analysis of the data (mean ± SEM) shows elevated protein expression expressed as mean fluorescence intensity (MFI) of (**G**) IL-8 and (**H**) MCP-1 in the cells co-stimulated with TNF-α and H_2_O_2_ as compared to those stimulated with TNF-α or H_2_O_2_ alone (*p* < 0.0001).

**Figure 4 ijms-22-10519-f004:**
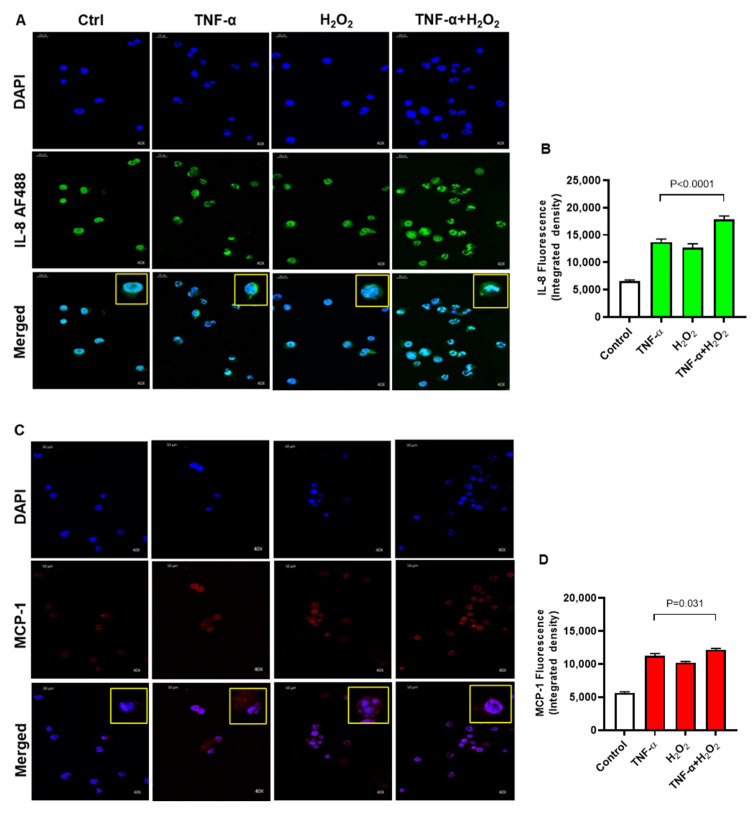
IL-8 and MCP-1 protein expression were determined by confocal microscopy. THP-1 cells were stimulated, in triplicate, with TNF-α (10 ng/mL) and/or H_2_O_2_ (10 mM) for 24 h, while control cells were treated with vehicle only. IL-8 and MCP-1 protein expression (fluorescence assessed as integrated density) was determined by confocal microscopy as detailed in materials and methods. For IL-8 and MCP-1 protein quantification, 10 random images (40× magnification) were captured for each treatment. The fluorescent channels were split using image J software (NIH, USA). The fluorescence intensity of IL-8 and MCP-1 was detected in 10 fields of each image and mean integrated density was calculated for each image. One-way ANOVA (Tukey’s multiple comparisons test) was used to calculate group differences and *p*-values less than 0.05 were considered as significant. The representative photomicrography images (40× magnification) obtained from three independent determinations with similar results show elevated expression of (**A**,**B**) IL-8 (*p* < 0.0001) and (**C**,**D**) MCP-1 (*p* = 0.031) in the cells co-stimulated with TNF-α and H_2_O_2_ as compared with cells stimulated with TNF-α or H_2_O_2_ alone.

**Figure 5 ijms-22-10519-f005:**
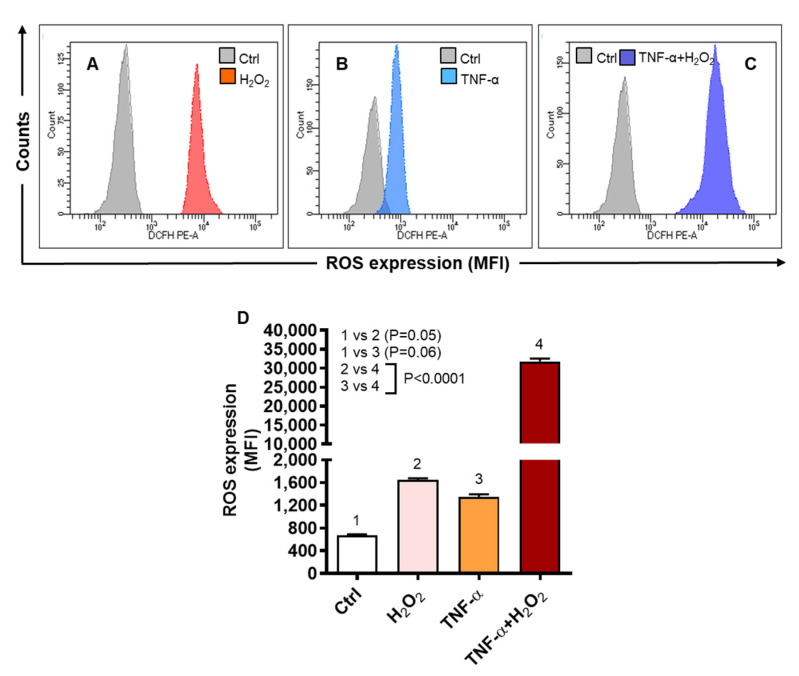
TNF-α/H_2_O_2_ cooperativity promotes the intracellular ROS expression. THP-1 cells were stimulated, in triplicate, with TNF-α (10 ng/mL) and/or H_2_O_2_ (10 mM), while control cells were treated with vehicle only. ROS expression was determined by using DCFH-DA assay and DCF expression, shown as mean fluorescence intensity (MFI), was measured by flow cytometry as described in materials and methods. One-way ANOVA (Tukey’s multiple comparisons test) was used to calculate group differences and *p*-values less than 0.05 were considered as significant. The representative histograms from three independent experiments with similar results show (**A**–**C**) comparative ROS expression in cells stimulated with TNF-α and/or H_2_O_2_ versus control. (**D**) Quantitative analysis of flow cytometry data (MFI: mean ± SEM) shows significantly elevated ROS expression in THP-1 cells co-stimulated with TNF-α and H_2_O_2_, compared to stimulation with either TNF-α or H_2_O_2_ (*p* ˂ 0.0001).

**Figure 6 ijms-22-10519-f006:**
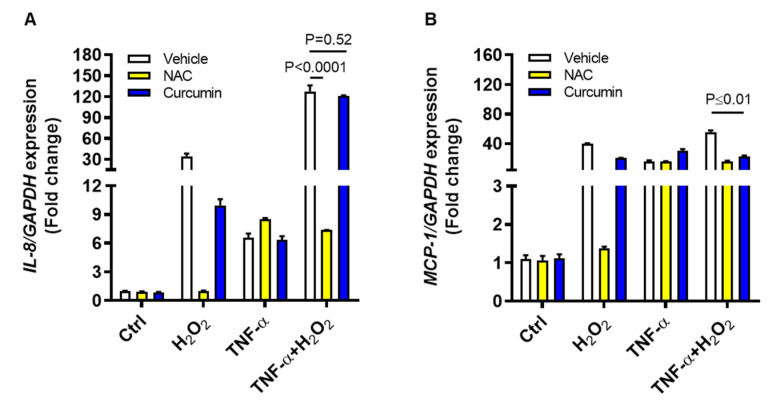
ROS inhibition suppresses the IL8 and MCP1 transcripts expression in monocytic cells co-stimulated with TNF-α and H_2_O_2_. THP-1 cells were stimulated, in triplicate, with TNF-α (10 ng/mL) and/or H_2_O_2_ (10 mM) while control cells were treated with vehicle only. In ROS inhibition assays, cells were preincubated for 30 min with either NAC (1 mM) or curcumin (10 μM) before stimulation with TNF-α and/or H_2_O_2_. Total RNA was extracted and IL8/MCP1 gene expression was assessed by qRT-PCR as described in methods. Two-way ANOVA was used to calculate group differences and *p*-values less than 0.05 were considered as significant. The representative data (mean ± SEM) obtained from three independent experiments with similar results show effective suppression of (**A**) IL8 mRNA by NAC (*p* ˂ 0.0001) and (**B**) MCP1 mRNA by both NAC and curcumin (*p* ≤ 0.01).

**Figure 7 ijms-22-10519-f007:**
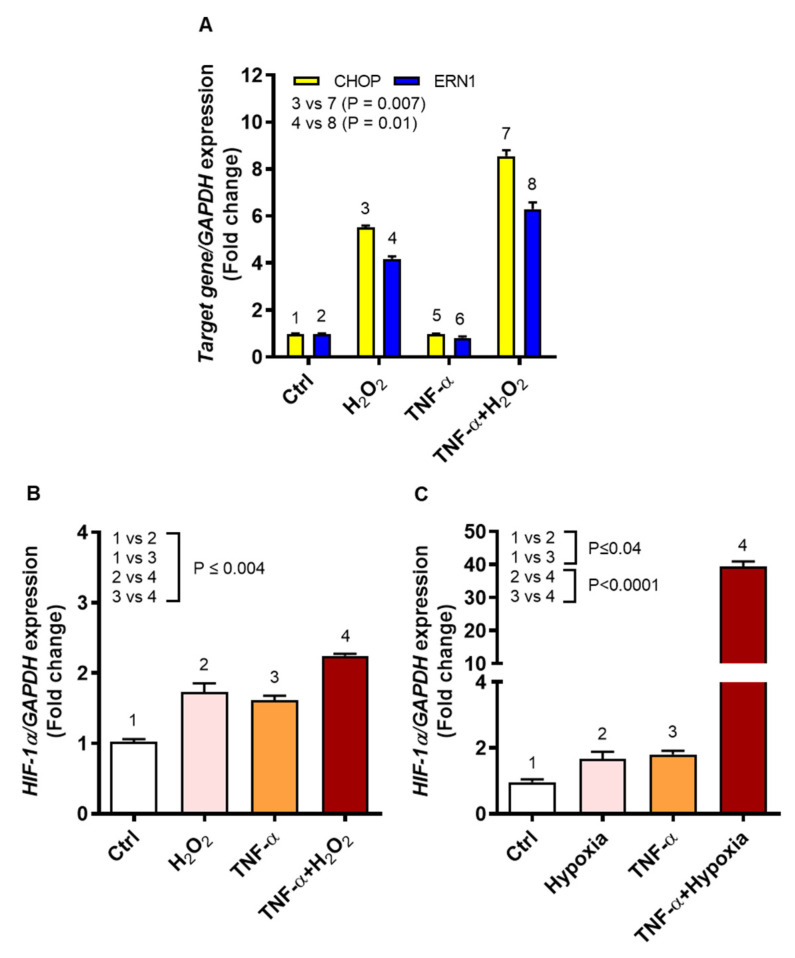
TNF-α/ROS cooperativity induces the ER stress and HIF1A stabilization. THP-1 cells were stimulated, in triplicate, with TNF-α (10 ng/mL), H_2_O_2_ (10 mM), and/or 1% hypoxia, while control cells were treated with vehicle only. Total RNA was purified for assessing gene expression of ER stress markers (CHOP and ERN1) and oxygen sensor (HIF1A) using qRT-PCR as detailed in methodology. As appropriate, one-way ANOVA (Tukey’s multiple comparisons test) or two-way ANOVA was used to calculate group differences and *p*-values less than 0.05 were considered as significant. The representative data (mean ± SEM) from two independent experiments with similar results show increased expression of: (**A**) CHOP and ERN1 in cells co-stimulated with TNF-α and H_2_O_2_ compared to cells treated with H_2_O_2_ alone (*p* ≤ 0.01); (**B**) HIF1A in cells co-stimulated with TNF-α and H_2_O_2_ compared to treatments with TNF-α or H_2_O_2_ alone (*p* ≤ 0.004); and (**C**) HIF1A in cells stimulated with TNF-α under hypoxia as compared to TNF-α treatment under normoxia or induction by hypoxia alone (*p* < 0.0001).

**Figure 8 ijms-22-10519-f008:**
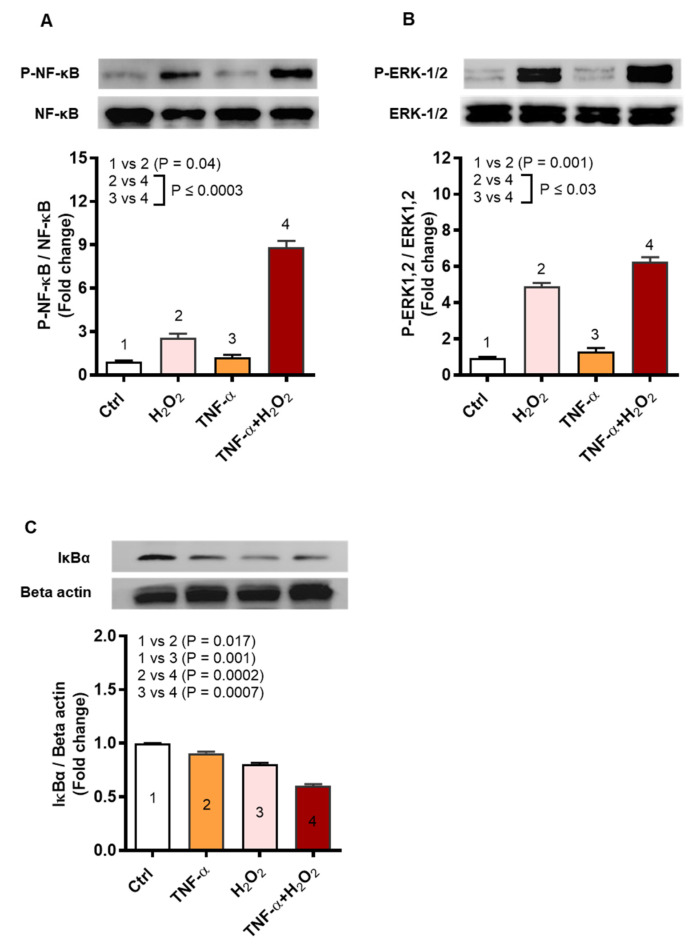
TNF-α/H_2_O_2_ cooperativity drives IL-8 and MCP-1 expression via activation of the NF-κB and ERK-1/2 mediated signaling. THP-1 monocytic cells were stimulated, in triplicate, with TNF-α (10 ng/mL) for 10 min and/or H_2_O_2_ (10 mM) for 30 min. Cell lysates were resolved by 12% SDS-PAGE and blots were probed with primary and secondary antibodies for detection of phosphorylated and total NF-κB (65 KDa) and ERK1/2 (42, 44 KDa), as well as for detection of total IκBα (36 KDa) and beta-actin (45 KDa). One-way ANOVA (Tukey’s multiple comparisons test) was used to calculate group differences and *p*-values less than 0.05 were considered as significant. Representative band densitometry data (mean ± SEM) from three independent experiments with similar results, following normalization show increased phosphorylation of (**A**) NF-κB (*p* ≤ 0.0003) and (**B**) ERK1/2 (*p* ≤ 0.03) as well as degradation of (**C**) IκBα (*p* ≤ 0.0007) in the cells co-stimulated with TNF-α and H_2_O_2_ as compared to those stimulated with TNF-α or H_2_O_2_ alone.

**Figure 9 ijms-22-10519-f009:**
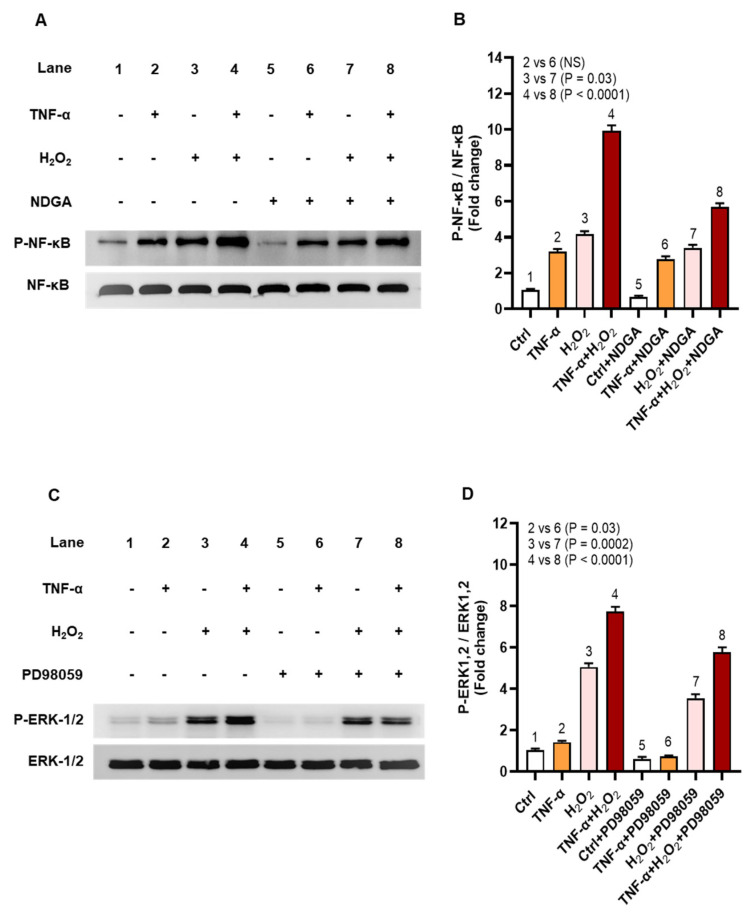
Chemical inhibition suppresses phosphorylation of NF-κB and ERK-1/2. To inhibit signaling via NF-κB and ERK-1/2 (MEK/ERK pathway), THP-1 cells were pre-incubated for 1 h, in triplicate wells, with nordihydroguaiaretic acid (NDGA; 30 μM) and PD98059 (10 μM), respectively, before stimulation with TNF-α (10 ng/mL) for 10 min and/or H_2_O_2_ (10 mM) for 30 min. Cell lysates were resolved by 12% SDS-PAGE and blots were probed with primary and secondary antibodies for detection of phosphorylated and total NF-κB (65 KDa) and ERK1/2 (42, 44 KDa). One-way ANOVA (Tukey’s multiple comparisons test) was used to calculate group differences and *p*-values less than 0.05 were considered as significant. Representative band densitometry data (mean ± SEM) from three independent experiments with similar results, following normalization show significantly reduced phosphorylation of (**A**,**B**) NF-κB and (**C**,**D**) ERK-1/2 in cells co-stimulated with TNF-α and H_2_O_2_ after treatment with pathway inhibitors compared to likewise stimulated cells without treatment with pathway inhibitors (*p* < 0.0001).

**Figure 10 ijms-22-10519-f010:**
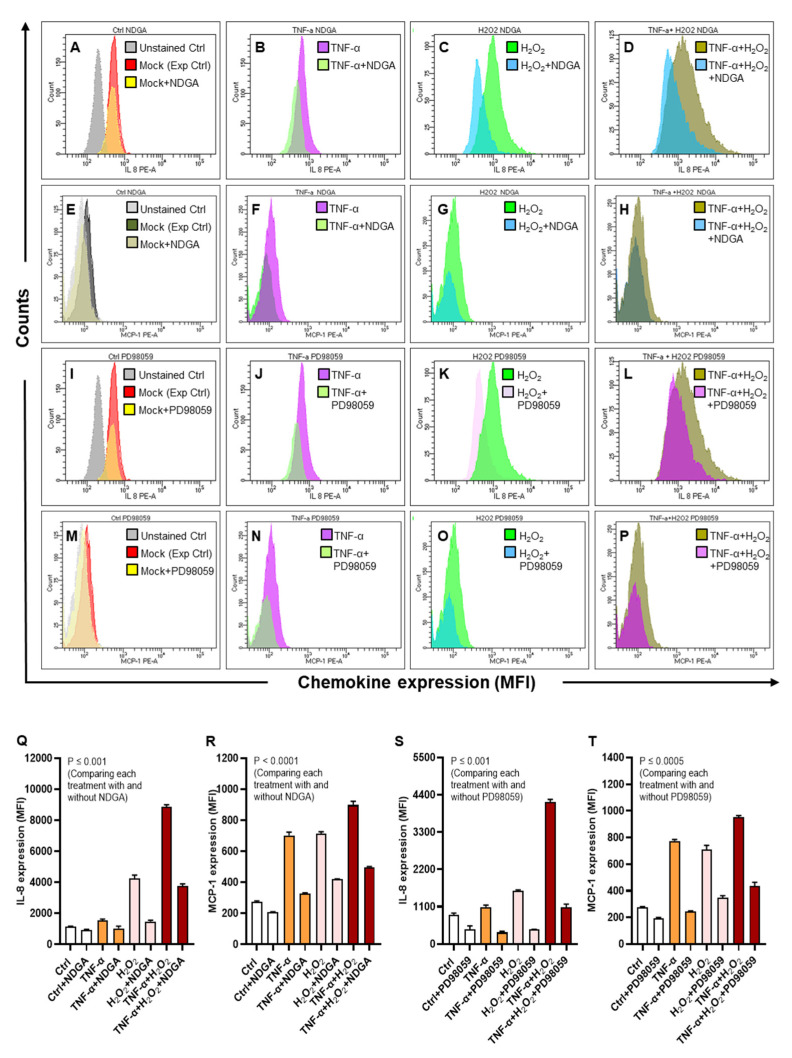
Inhibition of the NF-κB and ERK-1/2 activation suppresses the expression of IL-8 and MCP-1. THP-1 cells were pre-incubated for 1 h, in triplicate wells, with NF-κB inhibitor NDGA (30 μM) and ERK1/2 inhibitor PD98059 (10 μM), *TNF-α* (10. ng/mL) and/or H_2_O_2_ (10 mM) for 24 h, while control cells were treated with vehicle only. IL-8 and MCP-1 protein expression, expressed as mean fluorescence intensity (MFI), was measured using flow cytometry as described in materials and methods. One-way ANOVA (Tukey’s multiple comparisons test) was used to calculate group differences and *p*-values less than 0.05 were considered as significant. The representative histograms from three independent determinations with similar results show the comparative expression of (**A**–**D**) IL-8 and (**E**–**H**) MCP-1 in cells stimulated with TNF-α and/or H_2_O_2_, including control, with and without NDGA. Similarly, representative histograms from three independent determinations show the comparative expression of (**I**–**L**) IL-8 and (**M**–**P**) MCP-1 in cells stimulated with TNF-α and/or H_2_O_2_, including control, with and without PD98059. Quantitative analysis of the data (mean ± SEM) shows significant suppression of (**Q**) IL-8 (*p* ≤ 0.001) and (**R**) MCP-1 (*p* < 0.0001) in cells co-stimulated after pre-treatment with NDGA; as well as diminution of (**S**) IL-8 (*p* ≤ 0.001) and (**T**) MCP-1 (*p* ≤ 0.0005) in cells co-stimulated after pre-treatment with PD98059, as compared to respective controls without incubation with the pathway inhibitors.

**Figure 11 ijms-22-10519-f011:**
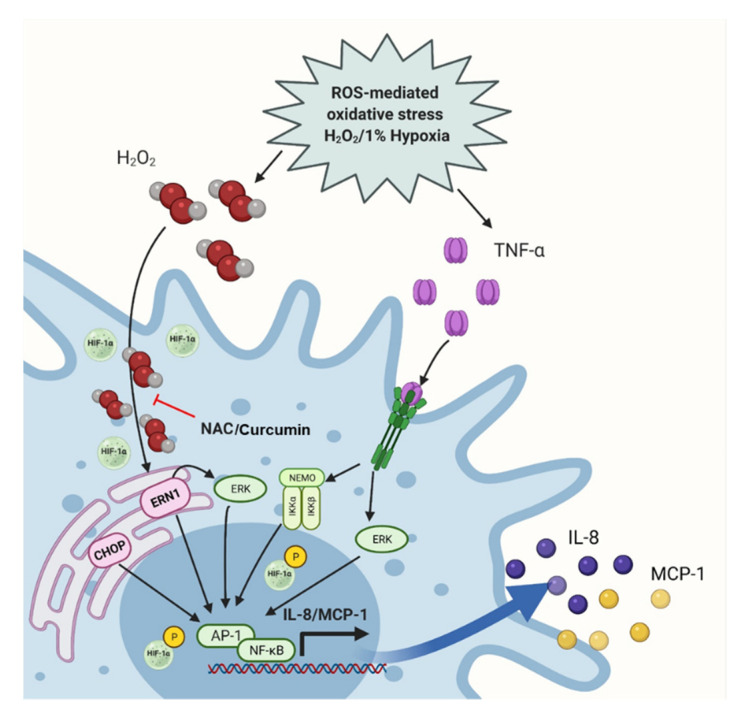
ROS/TNF-α crosstalk drives the IL-8 and MCP-1 expression. The schematic shows that THP-1 cell exposure to TNF-α and oxidative stress (induced by H_2_O_2_ or hypoxia) promotes the intracellular ROS together with *CHOP*, *ERN1*, and *HIF1A* upregulation, which leads to the NF-κB and ERK1-2 mediated expression of IL-8 and MCP-1; while the NAC or curcumin can counteract this chemokine induction. (Created with: BioRender.com; Accessed date: 18 March 2021).

## Data Availability

All data included in the study are presented in the manuscript. The corresponding author bears as guarantor for data validation.

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
