# Peer review of "ROS/TNF-α Crosstalk Triggers the Expression of IL-8 and MCP-1 in Human Monocytic THP-1 Cells via the NF-κB and ERK1/2 Mediated Signaling"

_ijms, 2021, doi:10.3390/ijms221910519_

Round 1
Reviewer 1 Report
The manuscript by N. Akhter et al. is not suitable for the publication in International Journal of Molecular Sciences.
The study lacks novelty and the link between the ex-vivo and in vitro section is not clear.
The results showed in the manuscript are not original as it is well known that adipose tissue of obese patients is inflamed compared to lean individuals as well as the induction of the inflammatory pathway by TNFalfa or H202 or both(indeed these methods are used as models of inflammation).
Thus the paper, in my opinion, do not have any new achievement so it does not deserve to be published.
Author Response
Point by point response to the comments by Reviewer 1
Comments and Suggestions for Authors
The manuscript by N. Akhter et al. is not suitable for the publication in International Journal of Molecular Sciences.
Comment 1: The study lacks novelty and the link between the ex-vivo and in vitro section is not clear.
Response 1: Thanks for your comments.
As of novelty concerns of the study, we would like to reiterate that the co-expression of both α- and β-chemokines, such as IL-8/CXCL8 and MCP-1/CCL2, respectively, has not been studied in macrophages, especially in the particular context of their amplification by the synergy between an inflammatory (TNF-α) and an oxidative stress (H2O2) inducer. However, induction of either IL-8 or MCP-1 in immune or other cell types by TNF-α is known. So, the novelty aspect of the study was to investigate the cooperativity or synergy between the inflammatory and the oxidative stress signals in driving induction of IL-8 and MCP-1 in monocytic cells, which might have significance in metabolic conditions, such as obesity.
As regards the link between ex-vivo and in-vitro sections of the study, the clinical data obtained using histochemistry show that both IL-8 and MCP-1 were elevated significantly in the adipose tissue samples from overweight and obese individuals which correlated directly with the levels of obesity (such as BMI). The confocal microscopy of adipose tissue samples further indicated that CD80+ M1 macrophages in obese tissue expressed these chemokines. Thus, the ex-vivo data of the study provide the premise for carrying out further investigations to address the study hypothesis that ROS-TNFα crosstalk may act as a chemokine (IL-8 and MCP-1) driver in the adipose tissue, especially in monocytic cells. The in-vitro data provide the evidence and establish proof of the concept that inflammatory and oxidative stress signals cooperatively generate the “synergistic effect” to trigger the IL-8 and MCP-1 expression in monocytic cells through the mechanisms involving ER stress, HIF1A stabilization, and NF-κB/ERK1-2 mediated signaling. The in-vitro data further show that NAC suppresses this chemokine amplification and may have therapeutic significance in meta-inflammatory setting.
Besides, in compliance to your previous comments (kindly refer to our previous rebuttal), we increased the number of clinical samples to address study strength concerns by you (clinical samples were increased from 15 to 45). Addressing your further concerns, we also performed new experiments and blots to show the IκBα degradation, in addition to showing increase in the phospho-IκBα / total-IκBα ratio. As well, addressing your concerns, new confocal microscopy stainings of the adipose tissue samples were done to show IL-8 and MCP-1 expression by the M1 macrophages in these samples. We have also used NF-κB inhibitor NDGA and ERK1/2 inhibitor PD98059 to confirm that synergistic induction of IL-8 and MCP-1 is mainly dependent on signaling via these two pathways. These data are added to the revised manuscript as new Fig 10.
To our knowledge, there is no study reported so far showing that ROS (oxidative stress stimulus) and TNF-α (inflammatory stimulus) synergy/ crosstalk leads to the amplification of IL-8 and MCP-1 expression in monocytic cells. However, the studies that have been cited in discussion section of the paper partially/ indirectly corroborate our findings, while these previous studies have also been done in contexts/ hypotheses different than ours.
Given all of the above, we hope that you will kindly consider all these efforts that we have made and new experiments that have been done along the course of the review process in compliance to your previous and present comments, concerns and suggestions.
Comment 2: The results showed in the manuscript are not original as it is well known that adipose tissue of obese patients is inflamed compared to lean individuals as well as the induction of the inflammatory pathway by TNF-alfa or H202 or both (indeed these methods are used as models of inflammation).
Thus the paper, in my opinion, do not have any new achievement so it does not deserve to be published.
Response 2: Thanks for your comment. As far as TNF-α is concerned, its increased adipose tissue expression in obesity is known, in accord with the increased inflammatory responses in white adipose tissue. However, the pathophysiological role of oxidative stress in the adipose tissue remains elusive and it is debated whether or not the presence of oxidative stress may exacerbate the inflammatory outcome in this compartment. It is true that as part of standard protocols, cells are treated with TNF-α to induce the inflammatory pathway, and with H2O2 to induce the ROS-associated oxidative stress. So, the use of these two agents in this study was to follow the standard protocol for co-inducing both the inflammatory and oxidative stress signals in monocytic cells, but it was a mean and not the end in itself, i.e. the actual purpose was to demonstrate how both stimuli synergistically drive the induction of IL-8 and MCP-1 in monocytic cells compared to when each stimulus was induced separately.
We hope that in light of all this explanation, you would feel convinced that the work done has enough originality in order to be published.
Submission Date
18 August 2021
Date of this review
30 Aug 2021 10:44:55
Reviewer 2 Report
In the present manuscript, Akhter and colleagues evaluated the potential interplay between TNF-α/oxidative stress and IL-8/MCP-1 in human THP-1 monocytes as well as in human subcutaneous adipose tissue from volunteers classified as normal weight, overweight or with obesity. The authors found an overexpression of IL-8 and MCP-1 in subcutaneous fat samples from patients with overweight and obesity. Moreover, THP-1 cells co-stimulated with TNF-α/H2O2 or TNF-α/hypoxia exhibited an increased mRNA and protein expression of IL-8 and MCP-1 through mechanisms involving endoplasmic reticulum stress, HIF-1α stabilization, and NFκB-/ERK1,2-mediated signalling. This is an interesting and well-designed study showing the relevance of oxidative stress on metabolic inflammation. Nonetheless, some specific points require to be amended.
Specific comments
- Figures: please remove the statistical analysis “One-way ANOVA…” and include it in the Figure legends.
- According to the approved gene nomenclature, human gene symbols generally are italicized, with all letters capitalized, without greek symbols and without hyphens (i.e. HIF1Afor the gene encoding HIF-1α). Human protein designations should be the same as the gene symbols except that all letters should be capitalized, in roman and with hyphens when necessary (i.e. HIF-1α). Please, change the human gene and protein symbols accordingly throughout the manuscript and figures.
- Line 40: please change “type-2 diabetes” to “type 2 diabetes”.
- Line 142 and throughout the Results’ section: Fold changes are usually explained as round numbers (e.g. please change “was increased 11.40±58 fold” to “was 11-fold increased”).
- Line 309: please change “fufther” to “further”.
- Please include a space between the number and its units (e.g. -80 ºC or 1 h), except from percentages (e.g. 5%).
- Line 687: please change the units of centrifugation in rpm to units of gravity (g) since rpm depends on the radius of the centrifuge.
Author Response
Point by point response to the comments by Reviewer 2
Comments and Suggestions for Authors
In the present manuscript, Akhter and colleagues evaluated the potential interplay between TNF-α/oxidative stress and IL-8/MCP-1 in human THP-1 monocytes as well as in human subcutaneous adipose tissue from volunteers classified as normal weight, overweight or with obesity. The authors found an overexpression of IL-8 and MCP-1 in subcutaneous fat samples from patients with overweight and obesity. Moreover, THP-1 cells co-stimulated with TNF-α/H2O2 or TNF-α/hypoxia exhibited an increased mRNA and protein expression of IL-8 and MCP-1 through mechanisms involving endoplasmic reticulum stress, HIF-1α stabilization, and NFκB-/ERK1,2-mediated signalling. This is an interesting and well-designed study showing the relevance of oxidative stress on metabolic inflammation. Nonetheless, some specific points require to be amended.
We thank the reviewer for kind comments. The point by point response to specific comments is given below:
Specific comments
Comment 1: Figures: please remove the statistical analysis “One-way ANOVA…” and include it in the Figure legends.
Response 1: Thanks for the comment. The concern has been addressed as suggested.
Comment 2: According to the approved gene nomenclature, human gene symbols generally are italicized, with all letters capitalized, without Greek symbols and without hyphens (i.e. HIF1A for the gene encoding HIF-1α). Human protein designations should be the same as the gene symbols except that all letters should be capitalized, in roman and with hyphens when necessary (i.e. HIF-1α). Please, change the human gene and protein symbols accordingly throughout the manuscript and figures.
Response 2: We thank the reviewer for the suggestion. The corrections have been made accordingly.
Comment 3: Line 40: please change “type-2 diabetes” to “type 2 diabetes”.
Response 3: Done as suggested.
Comment 4: Line 142 and throughout the Results’ section: Fold changes are usually explained as round numbers (e.g. please change “was increased 11.40±58 fold” to “was 11-fold increased”).
Response 4: Thanks. The modifications have been made as advised.
Comment 5: Line 309: please change “fufther” to “further”.
Response 5: Sorry for the typo. It is now corrected, and the whole document has been scanned using spellcheck.
Comment 6: Please include a space between the number and its units (e.g. -80 ºC or 1 h), except from percentages (e.g. 5%).
Response 6: Corrected as advised.
Comment 7: Line 687: please change the units of centrifugation in rpm to units of gravity (g) since rpm depends on the radius of the centrifuge.
Response 7: Thanks for pointing this out. The modification has been made as suggested.
Submission Date
18 August 2021
Date of this review
03 Sep 2021 18:13:37

This manuscript is a resubmission of an earlier submission. The following is a list of the peer review reports and author responses from that submission.
Round 1
Reviewer 1 Report
I find the revised manuscript to be improved and the points in my original critique to have largely been addressed. I recommend this manuscript for publication.
Author Response
Response to the comments by Reviewer 1
Comment 1: I find the revised manuscript to be improved and the points in my original critique to have largely been addressed. I recommend this manuscript for publication.
Response 1: The authors thank the worthy reviewer for kind approval of the MS revisions and for recommending it for publication.
Submission Date
04 May 2021
Date of this review
05 May 2021 13:46:25

Reviewer 2 Report
The work of Akhter et al. deals with an important and discussed the issue, that is obesity and inflammation. Particularly, the authors focus on analyzing the effect of TNFα and ROS combination in THP1 cells. However, the work lacks clarity, both from a stylistic and conceptual point of view and therefore needs a major revision.
MAJOR POINTS:
1-Which are the cells that express MCP-1 and IL-8 in adipose tissue? in Fig. S2 is represented CD163 but the data is not described.
2-Does the intensity of expression measured as staining intensity refer to the average of the observed fields or to the area of the
entire tissue section? How many fields have been observed? The number must be reported in materials and methods section.
3-It is well known that visceral tissue is the main site of inflammation. Why did you use subcutaneous adipose tissue sample?
4-The mechanism of hypoxia and oxidative stress are certainly linked but distinct. Could you better explain why you introduced the two mechanisms and why you chose to continue with the oxidative stress only?
The results in this way are not clear. Hypoxia should be better described.
5-As the authors explain well in the introduction, Il-8 and MCP-1 are cytokines that are secreted.
The gene expression and protein level data is fine but should be accompanied by an ELISA anlysis. 6-In figure 8 A and B the authors show that TNFa does not induce the phosphorylation of NF-kB and ERK.This data is in contrast with the literature. How do you explain it?
7-The authors write about synergy between TNFα and H2O2. Even if the synergy seems evident, the data must be supported by a specific synergy formula that you can find in the literature.
8-The English language must be reviewed by a native speaker.
MINOR:
1-It is necessary to add a table describing the population from which the adipose tissue samples are derived (sex, age, BMI, triglycerides, glycaemia, cholesterol ...)
2-Graphs need to be improved by a specific color panel that remains the same for each experimental condition, in order to improve the understanding of the images.
Author Response
Response to the comments by Reviewer 2
The work of Akhter et al. deals with an important and discussed the issue, that is obesity and inflammation. Particularly, the authors focus on analyzing the effect of TNFα and ROS combination in THP1 cells. However, the work lacks clarity, both from a stylistic and conceptual point of view and therefore needs a major revision.
MAJOR POINTS:
Comment 1: Which are the cells that express MCP-1 and IL-8 in adipose tissue? in Fig. S2 is represented CD163 but the data is not described.
Response 1: We thank for the comment. MCP-1 was expressed by monocytes/macrophages, and IL8 was found to be expressed by both adipocytes (data not shown) and monocytes/macrophages. In Fig S2, CD163 staining was used to show the co-localization of IL-8 expression on CD163+ cells (monocytes/macrophages) in the adipose tissue. Accordingly, the legend to Fig S2 has been now revised.
Comment 2: Does the intensity of expression measured as staining intensity refer to the average of the observed fields or to the area of the entire tissue section? How many fields have been observed? The number must be reported in materials and methods section.
Response 2: Thanks for the comment. The staining intensity measured represents the average of the observed fields. In this regard, 6 random fields (at 20× magnification) were selected from the entire tissue section. This information has now been corrected in Materials and Methods accordingly.
Comment 3: It is well known that visceral tissue is the main site of inflammation. Why did you use subcutaneous adipose tissue sample?
Response 3: We thank the reviewer for bringing this up. Further in this regard, although both subcutaneous adipose tissue (SAT) and visceral adipose tissue (VAT) express the identical set of genes, the level of expression is different in both adipose compartments. It has been reported that SAT and VAT are both associated with adverse cardiometabolic risk factors, however, the VAT remains more strongly associated with these risk factors [1]. On the other hand, it was reported by others that the expression of critical proinflammatory genes was remarkably higher in SAT than in VAT in individuals with morbid obesity [2]. Likewise, Spoto et al. have also reported that abdominal SAT contributed more than VAT to the proinflammatory milieu related to severe obesity [3]. In any case, our future studies will also include the abdominal VAT for further investigations as the available literature remains controversial.
Comment 4: The mechanism of hypoxia and oxidative stress are certainly linked but distinct. Could you better explain why you introduced the two mechanisms and why you chose to continue with the oxidative stress only? The results in this way are not clear. Hypoxia should be better described.
Response 4: Under normal cellular conditions, there is a homeostatic regulation or balance between the formation of reactive oxygen species (ROS) and their continuous elimination by the antioxidant enzymes such as superoxide dismutase, catalase, and glutathione peroxidase. It may be noted that abnormally low ROS levels can impair normal cellular signaling and some intracellular reactions; while, the elevated ROS levels induce oxidative stress and cause oxidation of lipids, proteins and DNA, resulting in altered expression of critical genes and pathways and eventually to cellular damage and death. Notably, a state of oxidative stress, marked by elevation of intracellular ROS, can be induced physically (as by hypoxia), or chemically (as by treatment with H2O2). In our study, since we found similar patterns of intracellular ROS elevation by using hypoxia and H2O2, most experiments were conducted using H2O2 which is a standard method of ROS induction in monocytic cells [4], without requiring lengthy incubations under limited oxygen conditions (involving a complex gaseous mixture of 1% O2, 5% CO2, and 94% N2) and use of specific incubator/ gaseous chamber. The inclusion of hypoxia data in our study shows that there are similar changes in the expression of the genes of interest (IL-8 and MCP-1), no matter whether oxidative stress is induced chemically (by H2O2 treatment) or physically (by 1% hypoxia).
Comment 5: As the authors explain well in the introduction, Il-8 and MCP-1 are cytokines that are secreted. The gene expression and protein level data is fine but should be accompanied by an ELISA analysis.
Response 5: Regarding this comment, since we already show conformity between the gene and protein expression of IL-8 and MCP-1 in monocytic cells, using qRT-PCR and FACS/IHC, respectively; we speculate it would have been superfluous to further use ELISA or Western blotting for assessing protein expression. Besides, the focal point was to demonstrate IL-8/MCP-1 protein expression in the cells following various treatments. The available literature supports the secretion aspect already (as we referred to in the introduction part). Under the prevailing COVID-19 circumstances, our institute is still working with less than 50% of its employees and ordering/receiving procedures of new IL-8 and MCP-1 ELISA kits may easily take another 10-16 weeks. The reviewer is, therefore, requested to kindly consider this constraint at the moment and oblige.
Comment 6: In figure 8 A and B the authors show that TNFa does not induce the phosphorylation of NF-kB and ERK. This data is in contrast with the literature. How do you explain it?
Response 6: Thanks for the comment. In our hands, although phosphorylation of NF-κB/ERK was induced (compared to control) in THP-1 cells in response to TNF-α treatment using a stimulatory pulse of 10 ng/mL for 10 min, however, it did not reach to the level of statistical significance, which may be due to inter-assay variation concerning the western blotting. As well, other studies [5-7] have used different cell types, concentrations, and/or exposure timings, all of which are likely to affect phosphorylation levels of NF-κB and ERK1/2.
Comment 7: The authors write about synergy between TNFα and H2O2. Even if the synergy seems evident, the data must be supported by a specific synergy formula that you can find in the literature.
Response 7: We thank the author for the comment. As though a synergistic interaction between oxidative stress and TNF-α stimulation is evident from the upregulation (at the transcriptional and translational levels) of both IL-8 and MCP-1 in THP-1 monocytic cells, we have replaced “synergy” with “cooperativity” throughout the revised manuscript version for the sake of better clarity.
Comment 8: The English language must be reviewed by a native speaker.
Response 8: In addressing this comment, the revised MS has been reviewed/edited by a native English speaker.
MINOR:
Comment 9: It is necessary to add a table describing the population from which the adipose tissue samples are derived (sex, age, BMI, triglycerides, glycaemia, cholesterol ...)
Response 9: Please see the supplementary Table S1 included in the revised MS version.
Comment 10: Graphs need to be improved by a specific color panel that remains the same for each experimental condition, in order to improve the understanding of the images.
Response 10: Done as suggested.
Submission Date
04 May 2021
Date of this review
25 May 2021 12:35:04
References
- Liu, J.; Fox, C. S.; Hickson, D. A.; May, W. D.; Hairston, K. G.; Carr, J. J.; Taylor, H. A., Impact of abdominal visceral and subcutaneous adipose tissue on cardiometabolic risk factors: the Jackson Heart Study. The Journal of clinical endocrinology and metabolism 2010, 95, (12), 5419-26.
- Mittal, B., Subcutaneous adipose tissue & visceral adipose tissue. The Indian journal of medical research 2019, 149, (5), 571-573.
- Spoto, B.; Di Betta, E.; Mattace-Raso, F.; Sijbrands, E.; Vilardi, A.; Parlongo, R. M.; Pizzini, P.; Pisano, A.; Vermi, W.; Testa, A.; Cutrupi, S.; D'Arrigo, G.; Lonardi, S.; Tripepi, G.; Cancarini, G.; Zoccali, C., Pro- and anti-inflammatory cytokine gene expression in subcutaneous and visceral fat in severe obesity. Nutr Metab Cardiovasc Dis 2014, 24, (10), 1137-43.
- Kim, D.; Kim, Y. J.; Koh, H. S.; Jang, T. Y.; Park, H. E.; Kim, J. Y., Reactive oxygen species enhance TLR10 expression in the human monocytic cell line THP-1. International journal of molecular sciences 2010, 11, (10), 3769-3782.
- Hop, H. T.; Reyes, A. W. B.; Huy, T. X. N.; Arayan, L. T.; Min, W.; Lee, H. J.; Rhee, M. H.; Chang, H. H.; Kim, S., Activation of NF-kB-Mediated TNF-Induced Antimicrobial Immunity Is Required for the Efficient Brucella abortus Clearance in RAW 264.7 Cells. Frontiers in Cellular and Infection Microbiology 2017, 7, (437).
- Sudhakar, C.; Nagabhushana, A.; Jain, N.; Swarup, G., NF-kappaB mediates tumor necrosis factor alpha-induced expression of optineurin, a negative regulator of NF-kappaB. PloS one 2009, 4, (4), e5114.
- Namba, S.; Nakano, R.; Kitanaka, T.; Kitanaka, N.; Nakayama, T.; Sugiya, H., ERK2 and JNK1 contribute to TNF-alpha-induced IL-8 expression in synovial fibroblasts. PloS one 2017, 12, (8), e0182923.

Reviewer 3 Report
The present manuscript adresses the role of NFκB- and ERK1/2-mediated signalling for TNF-α and oxidative stress induced expression of the chemokines IL-8/CXCL8 and MCP-1/CCL2 in the monocytic cell line THP-1. These pathways are relevant for the development of a pathophysiology in the context of obesity. In general, analyzing the molecular aspects of inflammation in this content is of interest as potential molecular targets for anti-oxidant or anti-inflammatory treatment need to be defined. However, the mechanistic aspects concerning the proposed signalling pathway presented by the authors are not new and anything but unclear.
It has already been suggested that TNF-α and H2O2 cooperate in the induction of the IL-8 promoter via NFκB and also AP-1 and in addition on the level of mRNA stability in epithelial cells and this has also a relevance for CCL2 expression (Roebuck, K.A. et al., 1999, J Leukoc Biol). Most likely similar events can be observed in THP-1 monocytes as there are several reports, which document that oxidative burst and TNF-α trigger IL-8 and CCL2 in this cell line and beside NFκB a role of ERK1/2 has also been proposed (e.g.: Ryan, K.A. et al., 2004, Infect Immunol; Chen, T. et al., 2011, Arch Dermatol Res.; Liu, J. et al., 2014, J Toxicol Environ Health; Mendez-Samperio, P., et al., 2010, Arch Med Res; Liu, Y.C. et al., 2017, Mol Immunol). Moreover, a role of HIF-1α in H2O2 induced CCL2 expression in THP-1 cells has been suggested, although not co-stimulated with TNF-α (Bai, W. et al., 2017, BBRC). Furthermore, it is state of the art that nutritionally mediated oxidative stress triggers inflammatory cascades via NFκB and as a result increased chemokine levels like those of IL-8 and CCL2 can be observed in obese humans (Munoz, A. and Costa, M., 2013, Oxid Med Cell Longev). The authors need to explain the novelity of their study in more detail and substantiate it with additional real new data.
Regarding the presented results in this study, the authors do not provide any evidence for the relevance of NFκB and ERK1/2 for TNF-α/H2O2 induced IL-8 or CCL2 expression. Demonstrating that these molecules are phosphorylated by H2O2 (figure 8) is not sufficient as there is no relation to IL-8 or CCL2 transcription. At least inhibitor or siRNA experiments with qPCR analysis for chemokine expression need be added. This is the weakest point in the manuscript. Therefore, the part of the title “...via NFκB and ERK1/2 mediated signaling” is not supported by respective results.
Nevertheless, the interesting point in this manuscript is the presentation of human samples from lean, overweight and obese individuals. Also the usage of different methods by the authors for analyzing IL-8 and CCL2 expression is commendable. The summarizing scheme is very nice, but the content is not new.
Author Response
Response to the comments by Reviewer 3
Comment 1: The present manuscript addresses the role of NFκB- and ERK1/2-mediated signalling for TNF-α and oxidative stress induced expression of the chemokines IL-8/CXCL8 and MCP-1/CCL2 in the monocytic cell line THP-1. These pathways are relevant for the development of a pathophysiology in the context of obesity. In general, analyzing the molecular aspects of inflammation in this content is of interest as potential molecular targets for anti-oxidant or anti-inflammatory treatment need to be defined. However, the mechanistic aspects concerning the proposed signalling pathway presented by the authors are not new and anything but unclear.
Response 1: We thank the reviewer for the valuable comments. In addressing these comments, please note that as though the involvement of NF-κB/ERK1-2 mediated signaling in chemokine expression is known, our data show for the first time the coherence between two insults, namely oxidative stress and immunostimulation by TNF-α, which drives the simultaneous expression of IL-8 (an α-chemokine) and MCP-1 (a β-chemokine) along with NF-κB/ERK1-2 hyperphosphorylation in THP-1 monocytic cells. Importantly, our data further show that ant-oxidants/ ROS-scavengers may have a potential to suppress such an inflammatory response.
Comment 2: It has already been suggested that TNF-α and H2O2 cooperate in the induction of the IL-8 promoter via NFκB and also AP-1 and in addition on the level of mRNA stability in epithelial cells and this has also a relevance for CCL2 expression (Roebuck, K.A. et al., 1999, J Leukoc Biol). Most likely similar events can be observed in THP-1 monocytes as there are several reports, which document that oxidative burst and TNF-α trigger IL-8 and CCL2 in this cell line and beside NFκB a role of ERK1/2 has also been proposed (e.g.: Ryan, K.A. et al., 2004, Infect Immunol; Chen, T. et al., 2011, Arch Dermatol Res.; Liu, J. et al., 2014, J Toxicol Environ Health; Mendez-Samperio, P., et al., 2010, Arch Med Res; Liu, Y.C. et al., 2017, Mol Immunol). Moreover, a role of HIF-1α in H2O2 induced CCL2 expression in THP-1 cells has been suggested, although not co-stimulated with TNF-α (Bai, W. et al., 2017, BBRC). Furthermore, it is state of the art that nutritionally mediated oxidative stress triggers inflammatory cascades via NFκB and as a result increased chemokine levels like those of IL-8 and CCL2 can be observed in obese humans (Munoz, A. and Costa, M., 2013, Oxid Med Cell Longev). The authors need to explain the novelty of their study in more detail and substantiate it with additional real new data.
Response 2: As though there are some reports in the available literature showing IL-8 or CCL2 (MCP-1) expression in different cells types in response to either TNF-α or H2O2, or even by nutritionally-mediated oxidative stress via NF-κB-mediated signaling, as show the studies cited by the worthy reviewer in the comment; however, the novelty of our work lies in the aspect that we demonstrate the cooperativity between oxidative stress and TNF-α stimulation which leads to the excessive expression of IL-8 and MCP-1 in monocytic cells. These preliminary findings pave way for further studies to investigate whether the agents alleviating oxidative stress could be beneficial in metabolic inflammatory settings.
Comment 3: Regarding the presented results in this study, the authors do not provide any evidence for the relevance of NFκB and ERK1/2 for TNF-α/H2O2 induced IL-8 or CCL2 expression. Demonstrating that these molecules are phosphorylated by H2O2 (figure 8) is not sufficient as there is no relation to IL-8 or CCL2 transcription. At least inhibitor or siRNA experiments with qPCR analysis for chemokine expression need be added. This is the weakest point in the manuscript. Therefore, the part of the title “...via NFκB and ERK1/2 mediated signaling” is not supported by respective results.
Response 3: We thank the reviewer for the comment. Our findings showing IL-8/MCP-1 expression via NF-κB/ERK-mediated signaling is in line with the previous evidences. Further, in this regard, Elliott et al. reported that the promoter region of the IL-8 gene had binding sites for NF-κB as well as other related transcription factors (AP-1 and C/EBP) [1]. Whereas, MCP-1 expression may involve both canonical [2] and non-canonical [3] pathways. Interestingly, Hoffmann et al. showed that excessive amounts of IL-8 were generated by a combination of three different mechanisms: (i) by de-repression of the IL-8 gene promoter; (ii) by transcriptional activation of the IL-8 gene by NF-κB & JUN-N-terminal protein kinase pathways; and (iii) by stabilization of the IL-8 mRNA by p38 MAPK pathway. The authors concluded that this multiple control mechanism enables cells rapidly increase as well as fine-tune the IL-8 expression [4].
However, we do agree with the worthy reviewer that inclusion of more data involving inhibitor or siRNA based experiments would have further substantiated the NF-κB/ERK dependence of IL-8 and MCP-1. Unfortunately, under the current pandemic situation, ordering and procuring these reagents would be problematic for us and take time like 3 or 4 months the minimum. Therefore, the authors request the reviewer to kindly grant a waiver and oblige.
Comment 4: Nevertheless, the interesting point in this manuscript is the presentation of human samples from lean, overweight and obese individuals. Also the usage of different methods by the authors for analyzing IL-8 and CCL2 expression is commendable. The summarizing scheme is very nice, but the content is not new.
Response 4: We thank the reviewer for the encouraging remarks. We are also aware that there is a limited novelty of the study, which we would strive to improve in our future studies on related aspects.
Submission Date
04 May 2021
Date of this review
25 May 2021 16:19:24
References
- Elliott, C. L.; Allport, V. C.; Loudon, J. A. Z.; Wu, G. D.; Bennett, P. R., Nuclear factor-kappa B is essential for up-regulation of interleukin-8 expression in human amnion and cervical epithelial cells. Molecular Human Reproduction 2001, 7, (8), 787-790.
- Rovin, B. H.; Dickerson, J. A.; Tan, L. C.; Hebert, C. A., Activation of nuclear factor-κB correlates with MCP-1 expression by human mesangial cells. Kidney international 1995, 48, (4), 1263-1271.
- Nakatsumi, H.; Matsumoto, M.; Nakayama, K. I., Noncanonical Pathway for Regulation of CCL2 Expression by an mTORC1-FOXK1 Axis Promotes Recruitment of Tumor-Associated Macrophages. Cell Rep 2017, 21, (9), 2471-2486.
- Hoffmann, E.; Dittrich-Breiholz, O.; Holtmann, H.; Kracht, M., Multiple control of interleukin-8 gene expression. Journal of leukocyte biology 2002, 72, (5), 847-855.

Round 2
Reviewer 2 Report
The manuscript by Akhter et al. has been slightly improved and the authors replied to all the questions.
Nevertheless, the paper lacks novelty and there are several unconvincing points.
For example, the number of histological samples is very low. Besides, the authors counted only 6 fields and in my opinion itis not sufficient.
Moreover, it is not commented why the authors used the cd163 marker, which is well known to be the marker of anti-inflammatory M2 macrophages and they observed the expression of IL8 cytokine.
Western blotting results are not convincing and not sufficient to explain the model you proposed.
I suggest increasing the number of samples and reconsidering the in vitro experiments if you want to publish in journals of this level.
Reviewer 3 Report
I do appreciate that the authors recognize that there is a limited novelity in their study and that they would consider this in their future studies. However, their responses did not address the critical points sufficiently as there are also experiments, which need to be done to substantiate the title of the manuscript. Amongst others, authors postulate in the title that the triggering is mediated by NF-κB and ERK1/2, but they still do not show any results to prove that. Refering other reports could be part of the discussion section, but it does not compensate deficits in the result section.